

# Branch point twist field form factors in the sine-Gordon model II: Composite twist fields and symmetry resolved entanglement

Dávid X. Horváth [1], Pasquale Calabrese [1,2] and Olalla A. Castro-Alvaredo [3]

**1** SISSA and INFN Sezione di Trieste, via Bonomea 265, 34136 Trieste, Italy
**2** International Centre for Theoretical Physics (ICTP), Strada Costiera 11, 34151 Trieste, Italy
**3** Department of Mathematics, City, University of London,
10 Northampton Square EC1V 0HB, UK

## Abstract

In this paper we continue the program initiated in Part I, that is the study of entanglement measures in the sine-Gordon model. In both parts, we have focussed on one specific technique, that is the well-known connection between branch point twist field correlators and measures of entanglement in 1+1D integrable quantum field theory. Our papers apply this technique for the first time to a non-diagonal theory with an involved particle spectrum, the sine-Gordon model. In this Part II we focus on a different entanglement measure, the symmetry resolved entanglement, and develop its associated twist field description, exploiting the underlying $U(1)$ symmetry of the theory. In this context, conventional branch point twist fields are no longer the fields required, but instead we must work with one of their composite generalisations, which can be understood as the field resulting from the fusion of a standard branch point twist field and the sine-Gordon exponential field associated with $U(1)$ symmetry. The resulting composite twist field has correlators which as usual admit a form factor expansion. In this paper we write the associated form factor equations and solve them for various examples in the breather sector by using the method of angular quantisation. We show that, in the attractive regime, this is the sector which provides the leading contribution to the symmetry resolved entropies, both Rényi and von Neumann. We compute the latter in the limit of a large region size and show that they satisfy the property of equipartition, that is the leading contribution to the symmetry resolved entanglement is independent of the symmetry sector.

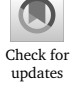

# 1 Introduction

The quantum relativistic sine-Gordon (sG) model is one of the most studied low-dimensional quantum field theories. This interest is justified by the fact that the sG model has many desirable features: it is a strongly correlated, genuinely interacting model with a rich particle spectrum, including topological excitations. Recently, due to spectacular advances in cold atom experiments, the sG model has been claimed to be a valid effective description of interacting Bose gases under certain circumstances [1, 2]. It is also a paradigmatic example of integrable quantum field theory (IQFT) and in particular, it provides the simplest example of an interacting theory with non-diagonal $S$-matrix [3]. Due the integrability the model is amenable to solution by the bootstrap program and consequently its exact mass spectrum, $S$-matrices and matrix elements of various fields are exactly known. In particular the particle spectrum contains two fundamental particles known as the soliton ($s$) and the antisoliton ($\bar{s}$) and a tower of breathers ($b_k$) which can be interpreted as bound states of solitons and antisolitons as well as bound states of lighter breathers. The number and masses of these breathers depend on the model's coupling constant. As already said the theory is non-diagonal in the standard scattering matrix sense as the scattering of solitons and antisolitons is generically accompanied by backscattering. Nevertheless the scattering in the breather sector is diagonal yielding considerable simplifications in many cases.

In the context of the bootstrap program for IQFTs, the matrix elements of local operators, known as form factors (FFs) of the sG model have been extensively studied by many authors employing many different techniques and in various contexts [4–20]. Particularly interesting operators are the branch point twist fields (BPTFs), whose form factors (FFs) in for the sG model were derived in [21,22]. The multi-point correlation functions of these fields, which are in principle calculable via their FFs, are directly related to entanglement entropies. Focusing on the simplest case, i.e., the bipartitioning of a quantum system in a pure state, the bipartite

entanglement of a subsystem $A$ can be quantified by the Rényi entropies [23–26]

$$S_n = \frac{1}{1-n} \ln \text{Tr} \rho_A^n, \tag{1}$$

defined in terms of the reduced density matrix (RDM) $\rho_A$ of the subsystem $A$. From Rényi entropies, in the so called replica limit $n \to 1$ the von Neumann entropy

$$S = -\text{Tr} \rho_A \ln \rho_A \tag{2}$$

is obtained, nevertheless the Rényi entropies for different $n$ contain more information than $S$, in particular their knowledge provides the entire spectrum of the RDM $\rho_A$ [27, 28].

Importantly, the Rényi and von Neumann entropies can be expressed in terms of certain partition functions as

$$S_n(\ell) = \frac{1}{1-n} \ln \left[ \frac{\mathcal{Z}_n}{\mathcal{Z}_1^n} \right], \qquad \text{and} \qquad S(\ell) = -\frac{\partial}{\partial n} \left[ \frac{\mathcal{Z}_n}{\mathcal{Z}_1^n} \right]_{n=1}, \tag{3}$$

where $\mathcal{Z}_n$ is the partition function of the theory on an $n$-sheeted Riemann surface $\mathcal{R}_n$, obtained by cyclically connecting the $n$ sheets along a subsystem $A$ [29–31]. Throughout this paper we take the subsystem to be a connected region of length $\ell$. In QFT the partition function $\mathcal{Z}_n$ can be shown to be proportional to a multi-point function of branch point twist fields, containing as many field insertions as boundary points between subsystem $A$ and the rest of the system [30, 31, 33]. These fields are local fields in a replica QFT, that is a new model consisting of $n$ non-interacting copies of the original theory. The BPTFs satisfies non-trivial exchange relations with other local fields of the theory [33]. Consequently, the moments of $\rho_A$ are generally equivalent to appropriate limits (i.e. for $n \to 1$) of multi-point functions of BPTFs.

The computation of these multi-point functions is, nevertheless, a very difficult task. Exact results including the scaling dimensions and multi-point functions of BPTFs are known in 2D conformal field theories (CFTs) [30, 34–44]. In 2D off-critical, mostly integrable and free theories, the form factor bootstrap allows for the calculation of the matrix elements of the BPTF, which has been carried out for many IQFTs besides the sG model [21, 33, 45–54]. Via the bootstrap program all matrix elements are computable, the multi-point correlation functions at large distances are, nevertheless, usually dominated by the first few sets of form factors. This property has been often exploited in IQFTs and the use of BPTFs and their FFs have resulted in numerous interesting exact or very accurate predictions for the entanglement entropy under many different physical circumstances [47–61].

Recently lot of attention was devoted to how internal symmetries of a particular model affect the structure of the reduced density matrix, hence the internal structure of entanglement measures. A new concept dubbed symmetry resolved entanglement (SRE) was introduced which can be studied using very similar techniques as reviewed above [62–65]. Consider a symmetry with an associated conserved charge $\hat{Q}$, which commutes with the system's Hamiltonian as well as with a density matrix of a symmetric state. Under general circumstances also the restriction of the symmetry operator $\hat{Q}$ to the subsystem denoted by $\hat{Q}_A$, commutes with the RDM, that is,

$$[\rho_A, \hat{Q}_A] = 0, \tag{4}$$

which has important consequences. First of all Eq. (4) implies that $\rho_A$ is block-diagonal and each block corresponds to an eigenvalue of $\hat{Q}_A$. Moreover this fact immediately indicates that the Rényi and von Neumann entropies can be decomposed according to the symmetry sectors of $\hat{Q}_A$. Accordingly one can define the symmetry resolved Rényi and von Neumann entropies as

$$S_n(q_A, \ell) = \frac{1}{1-n} \ln \left[ \frac{\mathcal{Z}_n(q_A)}{\mathcal{Z}_1^n(q_A)} \right], \qquad \text{and} \qquad S(q_A, \ell) = -\frac{\partial}{\partial n} \left[ \frac{\mathcal{Z}_n(q_A)}{\mathcal{Z}_1^n(q_A)} \right]_{n=1}, \tag{5}$$

via the symmetry resolved partition function

$$\mathcal{Z}_n(q_A) = \text{Tr}\left(\rho_A^n \mathcal{P}(q_A)\right), \tag{6}$$

where we denoted the projector onto the symmetry sector corresponding to the eigenvalue $q_A$ by $\mathcal{P}(q_A)$. Once more we write the dependence on the size of the system $\ell$ explicitly. SREs have been studied in various contexts such as conformal field theories (CFTs) [62, 63, 66–73], free [74,75] and interacting integrable QFTs [76], holographic settings [77], microscopic models on the lattice [62, 66–68, 78–86], out-of-equilibrium situations [67, 84, 87, 88] and for other systems exhibiting more exotic types of dynamics [89–94]. Notably, symmetry resolved quantities can be measured experimentally [87, 95].

Regarding the theoretical calculation of SREs, instead of directly computing the corresponding partition functions $\mathcal{Z}_n(q_A)$, it is more natural to consider the charged moments

$$Z_n(\alpha) = \text{Tr}\left(\rho_A^n e^{i\alpha \hat{Q}_A}\right), \tag{7}$$

which are Fourier transforms of the partition functions $\mathcal{Z}_n(q_A)$, so that [63]

$$\mathcal{Z}_n(q_A) = \text{Tr}\left(\rho_A^n \mathcal{P}(q_A)\right) = \int_{-\pi}^{\pi} \frac{d\alpha}{2\pi} Z_n(\alpha) e^{-i\alpha q_A}. \tag{8}$$

The charged moments have already been defined before their connection to symmetry resolution was established [96–102]. As pointed out in [63], they can be naturally interpreted as partition sums on the Riemann surface with an additional Aharonov-Bohm flux $e^{i\alpha}$ introduced on one of its sheets. Crucially, this picture can be rephrased for 1+1 dimensional field theories in a very similar manner to what was discussed; one can again consider multi-point functions of the composite branch point twist fields (CTF) in an $n$-copy theory. The boundary conditions along the cut(s) are now implemented by these novel fields, which also account for the insertion of the flux. These fields can be naturally regarded as a fusion of the standard BPTFs related to the permutation symmetry of the replicated model, and another twist field associated with the internal symmetry of the original theory.

In 2D CFT, the symmetry resolved entropies have been obtained as multi-point correlations of the novel CTFs [63]. These composite twist fields have been recently identified also in some massive theories: free massive Dirac and complex boson QFT [74,75], the off-critical Ising and sinh-Gordon theories [76]. In particular, as demonstrated in our previous works [75,76], FFs of the composite twist fields in IQFTs can be determined with the bootstrap program, similar to the usual BPTFs. Using these form factors and a systematic expansion for the correlation functions of the composite twist fields, symmetry resolved entropies can be computed.

As our title suggests, the aim of the present work is to further develop the program initiated in [22], extending it to CTFs with the aim of studying symmetry resolved quantities. In the sG model we will focus on the $U(1)$ symmetry associated with the the topological charge carried by the soliton/antisoliton. Accounting for this symmetry, we can write CTF form factor equations, as originally proposed in [76]. Concerning the solution of these equations, we focus as in [22], on the attractive regime of the sG model, where breathers are present in the spectrum. Indeed, in the present work we find solutions to the form factor bootstrap equations only for the breather sector.

This choice might seem limiting but is justified by the fact that, unlike for the standard BPTF studied in [22] where the first breather's one-particle form factor is zero by symmetry, in the present case the leading contribution of the charge moments is determined by the first breather. The solution of the bootstrap equations can actually be obtained by analytic continuation in the coupling from the sinh-Gordon (shG) model solution for an analogous field, whose validity can be easily checked using the $\Delta$-sum rule [103]. From this analytic continuation, we can easily

obtain the first breather FFs of the $U(1)$ CTF, as well as higher breathers FFs from the fusion procedure employed for instance in [22, 104, 105]. Finally we compute the $U(1)$ charged moments, when the subsystem is made of a single interval and the system is in the ground state, and from the charged moments we obtain the large-system leading contribution to the $U(1)$ symmetry resolved entropies. We confirm equipartitioning of entanglement at the leading order and compute the difference of the symmetry resolved and unresolved entropies as well, which depends not only the UV regulator $\varepsilon$ but also on the interaction strength of the sG model.

The paper is structured as follows: in section 2 we briefly review the sG and shG IQFTs and their main features. In section 3 the definitions and some important properties of the CTFs are listed. We also review some basic ingredients of IQFT in order to introduce a compact and convenient notation. Finally, we present the bootstrap equations for the $U(1)$ CTF in a generic IQFT and specialise them to the sG model. Section 4 is devoted to CTFs in the shG model. Besides presenting the first few FFs, non-trivial checks of these solutions are performed by taking specific limits in which we recover either the standard BPTF or the exponential field form factors. We also extensively employ the powerful $\Delta$-sum rule. Section 5 presents the one- and two-particle FFs of the $U(1)$ CTF in the sG model for the first few breathers. These are obtained by the above-mentioned analytic continuation and fusion procedures. In section 6 we first compute the $U(1)$ charged moments for an interval of length $\ell$ in the ground state of the sG model. We establish that the magnitude of leading large-distance correction to saturation is dominated by the one-particle form factor of the first breather. We then compute, via Fourier transform, the symmetry resolved entropies, focussing on their leading behaviour for large system size which exhibits the property of equipartition. We conclude in section 7. Some technical details are presented in appendices such as the determination of the shG FFs via the angular quantisation scheme in Appendix A and additional form factor checks via the $\Delta$-sum rule in Appendix B.

## 2 The Model

### 2.1 Sine-Gordon model

The sine-Gordon QFT is characterized by the following euclidean action

$$\mathcal{A} = \int d^2 x \left( \frac{1}{2} \partial_\mu \varphi \partial^\mu \varphi + \lambda \cos g \varphi \right), \tag{9}$$

where $\lambda$ is a coupling constants and $\varphi$ is a compactified scalar field, which means that the field is defined as

$$\varphi \equiv \varphi + \frac{2\pi}{g} k, \tag{10}$$

for any integer $k$. From the action above, the model admits an interpretation as massive perturbation of a compactified massless free boson, that is a CFT of central charge $c = 1$. This will play a role in later sections, when writing the conformal dimensions of various fields. The particle spectrum of the theory includes, first of all, a soliton $s$ and an antisoliton $\bar{s}$ forming a doublet of opposite $U(1)$ or topological charge. Depending on the value of $g$, we can distinguish the attractive and repulsive regime of the QFT. When $4\pi \le g^2 < 8\pi$, solitons and antisolitons repel each other, whereas if $g^2 < 4\pi$, they attract each other and consequently can form bound states. These bound states are called breathers and are topologically neutral

particles present in the asymptotic spectrum. Defining the new coupling strength

$$\xi = \frac{g^2}{8\pi - g^2}, \tag{11}$$

the number of different breather species is $\zeta(\xi) = [1/\xi]$, where $[\bullet]$ denotes the integer part. The value $g^2 = 4\pi$ corresponds to the free fermion point in the quantum theory; here the sG model is equivalent to the free massive Dirac fermion theory (as we can easily see from the $S$-matrices below), and the solitons and antisolitons correspond to the fundamental fermionic particle and antiparticle of the Dirac theory.

In the attractive regime, the masses of the breathers $m_k$ can be expressed in terms of the soliton mass $m$ as

$$m_k = 2m \sin \frac{\pi \xi k}{2}, \qquad \text{with} \qquad k = 1, \ldots, \zeta(\xi). \tag{12}$$

As is well known the theory is integrable and hence admits factorised scattering. The corresponding $S$-matrices have been determined explicitly [3]. Focusing first on the soliton sector of the sG model, the relevant S-matrices can be written as

$$
\begin{aligned}
S^{ss}_{ss}(\theta) = S^{\bar{s}\bar{s}}_{\bar{s}\bar{s}}(\theta) &= -\exp\left[ -i \int_0^\infty \frac{dt}{t} \frac{\sinh \frac{\pi t (1-\xi)}{2} \sin(t\theta)}{\sinh \frac{\pi t \xi}{2} \cosh \frac{\pi t}{2}} \right] \\
&= \prod_{k=0}^\infty \frac{\Gamma\left( \frac{2k+1}{\xi} - \frac{i\theta}{\pi\xi} + 1 \right) \Gamma\left( \frac{2k+1}{\xi} - \frac{i\theta}{\pi\xi} \right) \Gamma\left( \frac{2k}{\xi} + \frac{i\theta}{\pi\xi} + 1 \right) \Gamma\left( \frac{2k+2}{\xi} + \frac{i\theta}{\pi\xi} \right)}{\Gamma\left( \frac{2k}{\xi} - \frac{i\theta}{\pi\xi} + 1 \right) \Gamma\left( \frac{2k+2}{\xi} - \frac{i\theta}{\pi\xi} \right) \Gamma\left( \frac{2k+1}{\xi} + \frac{i\theta}{\pi\xi} \right) \Gamma\left( \frac{2k+1}{\xi} + \frac{i\theta}{\pi\xi} + 1 \right)},
\end{aligned}
\tag{13}
$$

and

$$S^{s\bar{s}}_{s\bar{s}}(\theta) = S^{\bar{s}s}_{\bar{s}s}(\theta) = \frac{\sinh \frac{\theta}{\xi}}{\sinh \frac{i\pi - \theta}{\xi}} S^{ss}_{ss}(\theta), \qquad S^{s\bar{s}}_{\bar{s}s}(\theta) = S^{\bar{s}s}_{s\bar{s}}(\theta) = \frac{\sinh \frac{i\pi}{\xi}}{\sinh \frac{i\pi - \theta}{\xi}} S^{ss}_{ss}(\theta), \tag{14}$$

where $S^{s\bar{s}}_{\bar{s}s}(\theta)$ and $S^{\bar{s}s}_{s\bar{s}}(\theta)$ are the off-diagonal amplitudes which can be seen to vanish whenever $1/\xi$ is an integer. Such values are known as reflectionless points. The remaining $S$-matrices are diagonal and can be expressed via the standard blocks:

$$[x]_\theta = \frac{\tanh \frac{1}{2}(\theta + i\pi x)}{\tanh \frac{1}{2}(\theta - i\pi x)}. \tag{15}$$

In particular

$$S_{b_1 b_1}(\theta) = [\xi]_\theta, \quad S_{b_2 b_2}(\theta) = [\xi]_\theta^2 [2\xi]_\theta, \tag{16}$$

$$S_{b_1 b_3}(\theta) = [\xi]_\theta [2\xi]_\theta, \quad S_{b_1 b_2}(\theta) = \left[\frac{\xi}{2}\right]_\theta \left[\frac{3\xi}{2}\right]_\theta, \tag{17}$$

where $b_k$ denotes the $k$th breather according the masses (12). These $S$-matrices have the important property of possessing poles in the physical sheet which one can attribute to the presence of a bound state. Similar to our previous work [22] the residue of such poles plays a role in later sections therefore we report some of these results here. In general, we define

$$-i \operatorname*{Res}_{\theta = i\pi u^c_{ab}} S_{ab}(\theta) := (\Gamma^c_{ab})^2, \tag{18}$$

where $i\pi u^c_{ab}$ is the pole of the $S$-matrix corresponding to the formation of a bound state $c$ in the scattering process $a + b \mapsto c$. Based on this equation a definition of the "pole strength" $\Gamma^c_{ab}$ can be naturally obtained. For the breather $S$-matrices, we have

$$\Gamma^{b_2}_{b_1 b_1} = \sqrt{2 \tan \pi \xi}, \quad \Gamma^{b_4}_{b_2 b_2} = \frac{2\cos \pi \xi + 1}{2\cos \pi \xi - 1} \sqrt{2 \tan 2\pi \xi}, \quad \Gamma^{b_3}_{b_1 b_2} = \sqrt{\frac{2\cos \pi \xi + 1}{2\cos \pi \xi - 1}} \, \Gamma^{b_2}_{b_1 b_1}, \tag{19}$$

and $\Gamma^{b_4}_{b_1 b_3} = \Gamma^{b_4}_{b_2 b_2}/\Gamma^{b_3}_{b_1 b_2}$. Finally we note that the breather $S$-matrices can also be expressed with the following integral formulae as

$$S_{b_k b_p}(\theta) = \exp\left[-i \int_0^\infty \frac{dt}{t} \frac{4\cosh\frac{\pi t\xi}{2}\sinh\frac{\pi t k\xi}{2}\cosh\frac{\pi t(1-\xi p)}{2}\sin(t\theta)}{\sinh\frac{\pi\xi t}{2}\cosh\frac{\pi t}{2}}\right], \qquad (20)$$

for $k < p$ and, finally

$$S_{b_k b_k}(\theta) = -\exp\left[-i \int_0^\infty \frac{dt}{t} \frac{2\left[\cosh\frac{\pi t\xi}{2}\sinh\frac{\pi t(2k\xi-1)}{2} + \sinh\frac{(1-\xi)\pi t}{2}\right]\sin(t\theta)}{\sinh\frac{\pi\xi t}{2}\cosh\frac{\pi t}{2}}\right]. \qquad (21)$$

Other $S$-matrices (like for $s - b_k$ scattering) as well as the derivation of Gamma-function representations from integral representations can be found for instance in [6].

## 2.2 Sinh-Gordon model

The sinh-Gordon model is defined by the action

$$\mathcal{A} = \int d^2x \left(\frac{1}{2}\partial_\mu\varphi\partial^\mu\varphi + \frac{\mu_0^2}{g^2}\cosh g\varphi\right), \qquad (22)$$

where $\mu_0$ is the bare particle mass, $g$ is the coupling constant and $\varphi$ is a non-compact bosonic scalar field. As in the sine-Gordon case, the sinh-Gordon model may be viewing as a massive perturbation of a massless free boson, albeit non-compactified in this case. Therefore, also in this case, the central charge of the underlying CFT is $c = 1$. The model is probably the simplest example of an interacting IQFT. The model is invariant under $\mathbb{Z}_2$ symmetry $\varphi \to -\varphi$ and its spectrum consists of multi-particle states of a single massive bosonic particle with exact mass $m$. The two-particle $S$-matrix is [109]

$$S(\theta) = \frac{\tanh\frac{1}{2}(\theta - i\frac{\pi B}{2})}{\tanh\frac{1}{2}(\theta + i\frac{\pi B}{2})}, \qquad (23)$$

where $B$ is related to the coupling $g$ in (22) by

$$B = \frac{2g^2}{8\pi + g^2}. \qquad (24)$$

As we can see, this is exactly $-2\xi$ under the replacement $g \mapsto ig$. Indeed, we have deliberately used the same notation $g$ for the coupling as we did for sG to emphasise the relationship between the two models. The action (22) can be formally obtained from the sine-Gordon action (9) under the analytic continuation $g \leftrightarrow ig$ but this applies to many other quantities too. This analytic continuation is a very useful tool for relating all sort of quantities in the shG model to those of the sG model, especially as the shG model is a much simpler theory where quantities such as form factors are more easily accessible. Under this correspondence, the shG $S$-matrix $S(\theta)$ and the sG first breather- first breather scattering matrix $S_{b_1 b_1}(\theta)$ can be obtained from one another. The same holds for the matrix elements of various operators, that is, the form factors which we study later.

# 3 Twist Fields in Quantum Field Theory

Twist fields are a common feature in quantum field theory and can be simply defined as fields associated to with an internal symmetry of the theory under consideration. In the context of

IQFT probably the simplest example is the order field $\sigma$ in Ising field theory, which is associated with discrete $\mathbb{Z}_2$ symmetry. It is in the nature of twist fields that they are non-local or semi-local with respect to other fields in the theory (i.e. the field $\sigma$ is semi-local with respect to the fermions in the Ising model) while at the same time being local in relation to the Lagrangian of the theory. Another standard feature of twist fields is that they sit at branch points in space-time, that is at the origin of branch cuts. From this feature, their semi-locality properties can be easily understood through exchange relations such as the ones shown later in this section. Going back to the Ising example above, the branch cut can be understood as resulting from the infinite sum over fermions that is involved in the Jordan-Wigner transformation that relates matrices $\sigma_\pm$ to fermions $c_i$ in the Ising spin chain whose scaling limit gives the Ising field theory. A good description of these features in the context of the FF program for the Ising model can be found in [106].

As discussed in Part I and originally in [33], an important twist field in the context of entanglement measures is the BPTF associated with cyclic permutation symmetry in a replica theory. Here, we want to focus on its generalization to a special type of composite BPTFs which we call composite twist fields (CTFs) for short. They are the fields that are formed by composition with another twist field. Note that composition with local (non twist) fields can also be considered, as done in [49,50] and this is of interest in the context of the entanglement entropy of non-unitary QFTs [52,53]. However, in this case the exchange relations and form factor equations are the same as for the original BPTF, that is, we are dealing with classifying multiple distinct solutions to the same FF equations. Here, once again the $\Delta$-sum rule plays an important role [103].

It is instructive to present the following formal definition of a CTF as given in [49]. Let $\mathcal{T}_n$ be the standard BPTF and $\phi$ be a local field in a conformal field theory, then the CTF

$$: \mathcal{T}_n \phi :(y) := n^{2\Delta-1} \lim_{x \to y} |x-y|^{2\Delta\left(1-\frac{1}{n}\right)} \sum_{j=1}^{n} \mathcal{T}_n(y)\phi_j(x), \tag{25}$$

may be defined, where $\phi_j(x)$ is the copy of field $\phi(x)$ living in replica $j$, $:\bullet:$ represents normal ordering and the power law, which involves the conformal dimension of the field $\phi$, denoted by $\Delta$. The conformal dimension of the BPTF is given by [34,35],

$$\Delta_n = \frac{c}{24}\left(n-\frac{1}{n}\right), \tag{26}$$

was derived in [49] from conformal arguments. The pre-factor $n^{2\Delta-1}$ ensures conformal normalization of the two-point function of CTFs. It is then natural to think of the CTFs we study in this paper as off-critical versions of their conformal counterparts (25).

As discussed in the introduction, for the study of symmetry resolved entanglement a special type of CTF is needed. This can be regarded as the composition of the standard BPTF with a twist field of the original (non-replicated) model. In the sG model there is a twist field associated with $U(1)$ symmetry, which can be represented as the simple vertex operator

$$\mathcal{V}_\alpha = \exp\left(\frac{i\alpha g\varphi}{2\pi}\right), \tag{27}$$

where $\varphi$ is the sG field and $g$ the coupling constant (see (9)). The exchange relations of this $U(1)$ twist field with other local fields in the theory are characterized by the semi-locality (or mutual locality) index $e^{i\kappa\alpha}$ as introduced in [106] via the equal time exchange relations

$$\mathcal{V}_\alpha(\mathbf{x})\phi_\kappa(\mathbf{y}) = e^{i\kappa\alpha}\phi_\kappa(\mathbf{y})\mathcal{V}_\alpha(\mathbf{x}) \quad \text{for} \quad y^1 > x^1, \tag{28}$$

$$= \phi_\kappa(\mathbf{y})\mathcal{V}_\alpha(\mathbf{x}) \qquad \text{for} \quad x^1 > y^1, \tag{29}$$

or, when using the radial quantisation picture,

$$\mathcal{V}_\alpha(e^{-2\pi i}z, e^{2\pi i}\bar{z})\phi_\kappa(0,0) = e^{i\kappa\alpha}\mathcal{V}_\alpha(z,\bar{z})\phi_\kappa(0,0), \tag{30}$$

with $\kappa = \pm, 0$. The interpolating fields $\phi_{\pm,0}$ are associated with the creation of a soliton (+), an antisoliton (−) or a neutral particle (0). It is important to stress that in the sG theory, the above exchange relations with the the precise phase factor $e^{i\kappa\alpha}$ are only recovered as long as the coupling constant $g$ appears in the definition the vertex operator (27). This fact can be best understood via semiclassical arguments: the $U(1)$ charge in the sG theory is associated with the solitons/antisolitons and a classical solitonic configuration, hence a charge unit is given by $2\pi/g$ increment in the classical field configuration. It is also well known, that the vertex operator is local w.r.t. fields $\phi_\pm$ creating solitons/antisolitons for $\alpha = 2\pi k$, $k \in \mathbb{Z}$.

The particular mutual locality factor $e^{i\kappa\alpha}$ and its physical meaning are in agreement with the intuitive picture associated with the insertion of the Aharonov-Bohm flux on one of the Riemann sheets. The picture with particles carrying the inserted flux is precisely rephrased by Eq. (30) in terms of quantum fields. Consequently, the $U(1)$ composite BPTF denoted as $\mathcal{T}_n^\alpha(x)$ can be understood formally as $:\mathcal{T}_n\mathcal{V}_\alpha:(x)$ in the sense of (25), and in a replica theory, is characterized by equal time exchange relations

$$\begin{aligned}
\mathcal{T}_n^\alpha(\mathbf{x})\mathcal{O}_{p,i}(\mathbf{y}) &= e^{\frac{ip\alpha}{n}}\mathcal{O}_{p,i+1}(\mathbf{y})\mathcal{T}_n^\alpha(\mathbf{x}) \quad \text{for} \quad y^1 > x^1, \tag{31}\\
&= \mathcal{O}_{p,i}(\mathbf{y})\mathcal{T}_n^\alpha(\mathbf{x}) \qquad \text{for} \quad x^1 > y^1, \tag{32}
\end{aligned}$$

with respect to quantum fields $\mathcal{O}_{p,i}$ living on the $i$th replica and possessing $U(1)$ charge $p \in \mathbb{Z}$. Similarly, if $\tilde{\mathcal{T}}_n(\mathbf{x})$ is the hermitian conjugate of $\mathcal{T}_n(\mathbf{x})$ associated with the inverse cyclic permutation, we can also define $\tilde{\mathcal{T}}_n^\alpha$ with exchange relations

$$\begin{aligned}
\tilde{\mathcal{T}}_n^\alpha(\mathbf{x})\mathcal{O}_{p,i}(\mathbf{y}) &= e^{-\frac{ip\alpha}{n}}\mathcal{O}_{p,i-1}(\mathbf{y})\tilde{\mathcal{T}}_n^\alpha(\mathbf{x}) \quad \text{for} \quad y^1 > x^1, \tag{33}\\
&= \mathcal{O}_{p,i}(\mathbf{y})\tilde{\mathcal{T}}_n^\alpha(\mathbf{x}) \qquad \text{for} \quad x^1 > y^1. \tag{34}
\end{aligned}$$

Our choice for $\pm\alpha/n$ in the exponents is motivated by the requirement, that the total phase picked up by a charged particle (associated with a unity of charge) has to be $e^{\pm i\alpha}$ when turning around each of the branch points. Based on the above relations and on previous works [75,76] we can easily write down the form factor bootstrap equations for the matrix elements of this CTF which we present in subsection 3.2. Before doing so we review some useful definitions and compact notations in IQFT.

## 3.1 Twist Field Form Factors in IQFT

For $\alpha = 0$ the exchange relations (31)-(33) become those of the standard BPTFs [33]. In IQFT one can then formulate BPTF form factor equations which generalize the standard form factor program for local fields [5,112]. These equations were first given in [33] for diagonal theories and then in [21] for non-diagonal ones. Here, we are interested in the further generalization to symmetry resolved CTFs, even if many elements of the derivation and notations are common.

Our most important object are the form factors (FF), which are matrix elements of (semi-)local operators $\mathcal{O}(x,t)$ between the vacuum and asymptotic states, i.e.,

$$F^{\mathcal{O}}_{\gamma_1\dots\gamma_k}(\theta_1,\dots,\theta_k) = \langle 0|\mathcal{O}(0,0)|\theta_1,\dots\theta_k\rangle_{\gamma_1\dots\gamma_k}. \tag{35}$$

In massive field theories like the sG model, the asymptotic states are spanned by multi-particle excitations whose dispersion relation can be parametrised as $(E,p) = (m_{\gamma_i}\cosh\theta, m_{\gamma_i}\sinh\theta)$, where $\gamma_i$ indicates the particle species and $\theta$ its rapidity. In such models, any multi-particle state can be constructed from the vacuum state $|0\rangle$ as

$$|\theta_1, \theta_2, ..., \theta_k\rangle_{\gamma_1\dots\gamma_k} = A^\dagger_{\gamma_1}(\theta_1)A^\dagger_{\gamma_2}(\theta_2)\dots A^\dagger_{\gamma_k}(\theta_k)|0\rangle, \tag{36}$$

where $A^\dagger$s are particle creation operators; in particular the operator $A^\dagger_{\gamma_i}(\theta_i)$ creates a particle of species $\gamma_i$ with rapidity $\theta_i$. In an IQFT with factorised scattering, the creation and annihilation operators $A^\dagger_{\gamma_i}(\theta)$ and $A_{\gamma_i}(\theta)$ satisfy the Zamolodchikov-Faddeev (ZF) algebra [107,108] which in the non-diagonal case reads

$$
\begin{aligned}
A^\dagger_{\gamma_i}(\theta_i)A^\dagger_{\gamma_j}(\theta_j) &= S^{\delta_i\delta_j}_{\gamma_i\gamma_j}(\theta_i-\theta_j)A^\dagger_{\delta_j}(\theta_j)A^\dagger_{\delta_i}(\theta_i)\,, \\
A_{\gamma_i}(\theta_i)A_{\gamma_j}(\theta_j) &= S^{\delta_i\delta_j}_{\gamma_i,\gamma_j}(\theta_i-\theta_j)A_{\delta_j}(\theta_j)A_{\delta_i}(\theta_i)\,, \\
A_{\gamma_i}(\theta_i)A^\dagger_{\gamma_j}(\theta_j) &= S^{\delta_i\delta_j}_{\gamma_i\gamma_j}(\theta_j-\theta_i)A^\dagger_{\gamma_j}(\theta_j)A_{\gamma_i}(\theta_i) + \delta_{\gamma_i,\gamma_j}2\pi\delta(\theta_i-\theta_j)\,,
\end{aligned}
\tag{37}
$$

where $S^{\delta_i\delta_j}_{\gamma_i\gamma_j}(\theta_i-\theta_j)$ denotes the two-body S-matrix of the theory and summation is understood on repeated indices. The above discussion is general and valid for any IQFT including the sG model, where the particle index $\gamma_i$ can take the particular values $b_1,\ldots,b_\zeta$ and $s,\bar{s}$.

In the $n$-copy IQFT each of the indices above is doubled, in the sense that particles are characterized both by their species ($\gamma_i$, $\delta_i$ in the formulae below) and their copy number ($\mu_i$, $\nu_i$ in the formulae below). The two-body scattering matrix is then generalized to

$$
S^{(\delta_i,\nu_i)(\delta_j,\nu_j)}_{(\gamma_i,\mu_i)(\gamma_j,\mu_j)}(\theta) = \delta_{\mu_i,\nu_i}\delta_{\mu_j,\nu_j}
\begin{cases}
S^{\delta_i\delta_j}_{\gamma_i\gamma_j}(\theta) & \mu_i=\mu_j \\
\delta_{\gamma_i,\delta_i}\delta_{\gamma_j,\delta_j} & \mu_i\neq\mu_j
\end{cases}.
\tag{38}
$$

To make our notations easier we introduce the multi-index

$$
a_i = (\gamma_i,\mu_i), \quad \text{with} \quad \bar{a}_i = (\bar{\gamma}_i,\mu_i) \quad \text{and} \quad \hat{a}_i = (\gamma_i,\mu_i+1),
\tag{39}
$$

where $\bar{\gamma}_i$ denotes the antiparticle of $\gamma_i$.

## 3.2 Form Factor Equations for U(1) CTFs

Relying on the exchange properties of the $U(1)$ CTFs (31) and also on earlier works [75,76] we can easily write down the bootstrap equations for the novel composite twist fields. Importantly, these equations include the non trivial phase $e^{\frac{i\alpha}{n}}$ in the monodromy properties corresponding the Aharonov-Bohm flux. Denoting the FFs of $\mathcal{T}^\alpha_n$ by $F^\alpha_{a_1\ldots a_k}(\theta_1,\ldots,\theta_k;\xi,n)$ (recall that $\xi$ is the sG coupling and $n$ the replica number), the bootstrap equations can be formulated as

$$
F^\alpha_{\underline{a}}(\underline{\theta};\xi,n) = S^{a'_i a'_{i+1}}_{a_i a_{i+1}}(\theta_{i\,i+1})F^\alpha_{\ldots a_{i-1}a'_{i+1}a'_i a_{i+2}\ldots}(\ldots\theta_{i+1},\theta_i,\ldots;\xi,n)\,,
\tag{40}
$$

$$
F^\alpha_{\underline{a}}(\theta_1+2\pi i,\theta_2,\ldots,\theta_k;\xi,n) = e^{\frac{i\kappa_1\alpha}{n}}F^\alpha_{a_2 a_3\ldots a_k\hat{a}_1}(\theta_2,\ldots,\theta_k,\theta_1;\xi,n)\,,
\tag{41}
$$

$$
-i\operatorname*{Res}_{\theta'_0=\theta_0+i\pi}F^\alpha_{\bar{a}_0 a_0\underline{a}}(\theta'_0,\theta_0,\underline{\theta};\xi,n) = F^\alpha_{\underline{a}}(\underline{\theta};\xi,n)\,,
\tag{42}
$$

$$
-i\operatorname*{Res}_{\theta'_0=\theta_0+i\pi}F^\alpha_{\bar{a}_0 \hat{a}_0\underline{a}}(\theta'_0,\theta_0,\underline{\theta};\xi,n) = -e^{\frac{i\kappa_0\alpha}{n}}\mathcal{S}^{\underline{a}'}_{\hat{a}_0\underline{a}}(\theta_0,\underline{\theta},k)F^\alpha_{\underline{a}'}(\underline{\theta};\xi,n)\,,
$$

$$
-i\operatorname*{Res}_{\theta'_0=\theta_0+i\bar{u}^\varepsilon_{\gamma\delta}}F^\alpha_{(\gamma,\mu_0)(\delta,\mu'_0)\underline{a}}(\theta'_0,\theta_0,\underline{\theta};\xi,n) = \delta_{\mu_0,\mu'_0}\Gamma^\varepsilon_{\gamma\delta}F^\alpha_{(\varepsilon,\mu_0)\underline{a}}(\theta_0,\underline{\theta};\xi,n)\,,
\tag{43}
$$

where several short-hand notations have been used: as usual $\theta_{ij}=\theta_i-\theta_j$, $\underline{\theta}:=\theta_1,\theta_2,...,\theta_k$ and $\underline{a}:=(\gamma_1,\mu_1)(\gamma_2,\mu_2)\ldots(\gamma_k,\mu_k)$. The factor in the fourth equation is an abbreviation for

$$
\mathcal{S}^{\underline{a}'}_{\hat{a}_0\underline{a}}(\theta_0,\underline{\theta},k) = S^{c_1 d_1}_{\hat{a}_0 a_1}(\theta_{01})S^{c_2 d_2}_{c_1 a_2}(\theta_{02})\ldots S^{\hat{a}_0 d_k}_{c_{k-1}a_k}(\theta_{0k})\,.
\tag{44}
$$

Note that the CTF is generally spinless and therefore the form factors are functions of rapidity differences only. The index $\kappa$ in the phase factors corresponds the $U(1)$ charge of the corresponding particle, that is

$$\kappa_i = \begin{cases} 1 & \gamma_i = s \\ -1 & \gamma_i = \bar{s} \\ 0 & \gamma_i = b_j \end{cases}, \tag{45}$$

which means that the non-trivial monodromy does not affect the breather sector of the theory. From this point we can proceed analogously to the previous case of the standard BPTFs; we primarily focus on one- and two-particle FFs and work on the first replica only. The one-particle FFs when non-vanishing, are again rapidity independent and the two-particle ones depend only on the rapidity difference. Akin to the BPTF, the novel composite field is neutral in relation to the sG $U(1)$-symmetry, which implies the vanishing of any FFs involving a different number of solitons and antisolitons. In particular one finds that

$$F_{ss}^{\alpha}(\theta;\xi,n) = F_{\bar{s}\bar{s}}^{\alpha}(\theta;\xi,n) = F_{\bar{s}b_k}^{\alpha}(\theta,\xi;n) = F_{sb_k}^{\alpha}(\theta,\xi;n) = F_s^{\alpha}(\xi,n) = F_{\bar{s}}^{\alpha}(\xi,n) = 0, \forall k \in \mathbb{Z}^+. \tag{46}$$

Unlike the BPTF, however, the $\mathbb{Z}_2$ symmetry imposes no additional restrictions and as an important consequence we have non vanishing one-particle and two-particle FFs for all breather combinations.

Under these considerations, Watson's equations (40), (41) for non-vanishing two-particle form factors and particles in the same copy can be summarised as

$$F_{s\bar{s}}^{\alpha}(\theta;\xi,n) = S_{s\bar{s}}^{s\bar{s}}(\theta)F_{\bar{s}s}^{\alpha}(-\theta;\xi,n) + S_{s\bar{s}}^{\bar{s}s}(\theta)F_{s\bar{s}}^{\alpha}(-\theta;\xi,n) = e^{i\alpha}F_{\bar{s}s}^{\alpha}(2\pi in - \theta;\xi,n), \tag{47}$$

$$F_{\bar{s}s}^{\alpha}(\theta;\xi,n) = S_{\bar{s}s}^{\bar{s}s}(\theta)F_{s\bar{s}}^{\alpha}(-\theta;\xi,n) + S_{\bar{s}s}^{s\bar{s}}(\theta)F_{\bar{s}s}^{\alpha}(-\theta;\xi,n) = e^{-i\alpha}F_{s\bar{s}}^{\alpha}(2\pi in - \theta;\xi,n), \tag{48}$$

$$F_{b_i b_j}^{\alpha}(\theta;\xi,n) = S_{b_i b_j}(\theta)F_{b_i b_j}^{\alpha}(-\theta;\xi,n) = F_{b_i b_j}^{\alpha}(2\pi in - \theta;\xi,n) \quad \text{,for} \quad i-j \in 2\mathbb{Z}. \tag{49}$$

The kinematic residue equations (42) are

$$-i\operatorname*{Res}_{\theta=i\pi} F_{s\bar{s}}^{\alpha}(\theta;\xi,n) = -i\operatorname*{Res}_{\theta=i\pi} F_{b_i b_i}^{\alpha}(\theta;\xi,n) = \langle \mathcal{T}_n^{\alpha} \rangle, \quad \forall \quad i \in \mathbb{N}, \tag{50}$$

where $\langle \mathcal{T}_n^{\alpha} \rangle$ is the vacuum expectation value of the CTF in the ground state of the replica theory. Finally, the bound state residue equations (43) are

$$-i\operatorname*{Res}_{\theta=i\pi u_{s\bar{s}}^c} F_{s\bar{s}}^{\alpha}(\theta;\xi,n) = \Gamma_{s\bar{s}}^c F_c^{\alpha}(\xi;n), \tag{51}$$

where $c$ is any particle that is formed as a bound state of $s+\bar{s}$ for rapidity difference $\theta = i\pi u_{s\bar{s}}^c$. For the breather sector it is again convenient to write the more general equation

$$-i\operatorname*{Res}_{\theta=\theta_0} F_{b_i,b_j,\underline{a}}^{\alpha}(\theta + iu, \theta_0 - i\tilde{u}, \underline{\theta};\xi,n) = \Gamma_{b_i b_j}^{b_{i+j}} F_{b_{i+j},\underline{a}}^{\alpha}(\theta, \underline{\theta};\xi,n), \tag{52}$$

where $\underline{a}$ is any particle combination for which the FF is non-vanishing. We recall that $u + \tilde{u} = u_{ij}^{i+j}$ and $\theta = i\pi u_{ij}^{i+j}$ is the pole of the scattering matrix $S_{b_i b_j}(\theta)$ and $u$ and $\tilde{u}$ are related to the poles of $S_{b_j b_{i+j}}(\theta)$ and $S_{b_i b_{i+j}}(\theta)$, respectively. It is important to emphasise that the bootstrap equations (40)-(43) or (49)-(51) for the $U(1)$ neutral breathers are identical to those of the conventional BPTFs, nevertheless the FFs are clearly different from those of $\mathcal{T}_n$ and their computation is non-trivial as demonstrated soon.

Finally we stress that two-particle FFs with arbitrary replica indices can be straightforwardly obtained from the above quantities (corresponding to particles on the same replica only) through the relations

$$F_{(\gamma,j)(\delta,k)}^{\alpha}(\theta;\xi,n) = \begin{cases} e^{i\alpha\kappa_{\delta}/n}F_{\delta\gamma}^{\alpha}(2\pi i(k-j)-\theta;\xi,n) & \text{if } k > j, \\ e^{i\alpha\kappa_{\gamma}/n}F_{\gamma\delta}^{\alpha}(2\pi i(j-k)+\theta;\xi,n) & \text{otherwise}, \end{cases} \tag{53}$$

where $\kappa_\gamma$ is zero for neutral breathers. The two-particle FFs of the other field $\tilde{\mathcal{T}}_n^{\ \alpha}$ denoted by $\tilde{F}_{a_1 a_2}^\alpha(\theta; \xi, n)$ can be simply written as [75]

$$\tilde{F}_{(\gamma,j)(\delta,k)}^\alpha(\theta; \xi, n) = F_{(\gamma,n-j)(\delta,n-k)}^{-\alpha}(\theta; \xi, n). \tag{54}$$

# 4 Form factors of the Exponential CTF in the sinh-Gordon model

As discussed in the Introduction, in this work we focus on the attractive regime of the sG model and restrict ourselves to the breather FFs of $\mathcal{T}_n^\alpha$. Given our goals, and in line with the strategy followed in Part I, we start by computing the FFs of the analogous field in the shG model. As this model has only one particle species, the powerful $\Delta$-sum rule (see Eq. (72)) can be used to verify the solutions for the composite field. With these solutions at hand one can then use the standard analytic continuation to obtain $b_1$ FFs of the $U(1)$ CTF in the sG theory and then use the fusion procedure just as in [22] to compute higher breather FFs.

As shown in [22], the multi-particle $b_1$ FFs of the standard BPTF in the sG theory can be obtained from the corresponding BPTF FFs of the shG model by identifying the shG coupling $B$ with $-2\xi$, where $\xi$ is the sG coupling defined in (11). The validity of the procedure was only shown for a few form factors but is supported by the general relationship between the lagrangians of the two theories and by a similar relation holding for other local fields, such as shG exponential fields $e^{\frac{\alpha g \varphi}{2\pi}}$. The validity of the fusion procedure and analytic continuation for the form factors of these fields was advocated in [12] and in general we have that

$$\text{FFs of} \quad e^{\frac{\alpha g \varphi}{2\pi}} \quad \text{in shG} \quad \mapsto \quad \text{FFs of} \quad e^{\frac{i\alpha g \varphi}{2\pi}} \quad \text{in sG}, \tag{55}$$

for $g \mapsto ig$ and $B \mapsto -2\xi$. In this section, we focus on computing the first few FFs of the exponential CTF in the shG model. We will use the notation

$$F_k^\alpha(\theta_1, \ldots, \theta_k; B, n) \equiv \text{ the FFs of } \mathcal{T}_n^\alpha =: \mathcal{T}_n e^{\frac{g \alpha \varphi}{2\pi}} : \text{ in shG}. \tag{56}$$

Therefore the FFs of the exponential field are a special case of the above, namely

$$F_k^\alpha(\theta_1, \ldots, \theta_k; B, 1) \equiv \text{ the FFs of } e^{\frac{\alpha g \varphi}{2\pi}} \text{ in shG}, \tag{57}$$

as are the form factors of the standard branch point twist field

$$F_k^0(\theta_1, \ldots, \theta_k; B, n) \equiv \text{ the FFs of } \mathcal{T}_n \text{ in shG}. \tag{58}$$

Here we briefly discuss first the determination of the two-particle FF using standard methods, from which the one-particle FF can be easily computed as well. The three- and four-particle FFs are presented in appendix A, where they are obtained by the method of angular quantisation [122]. To obtain the two-particle FF, we solve Watson's equations and the kinematic pole equation as usual

$$F_2^\alpha(\theta; B, n) = S(\theta) F_2^\alpha(-\theta; B, n) = F_2^\alpha(2\pi i n - \theta; B, n), \tag{59}$$

and

$$-i \operatorname*{Res}_{\theta = i\pi} F_2^\alpha(\theta; B, n) = \langle \mathcal{T}_n^\alpha \rangle, \tag{60}$$

where $S(\theta)$ is the $S$-matrix (23) and there are no bound states, so this the full set of FF equations. These equations are the same as for the standard BPTF but the solutions we are looking for must be different. In particular, they must depend on the parameter $\alpha$ so that when $\alpha = 0$ we should recover the known BPTF solutions [33, 48] and when $n = 1$ with $\alpha \neq 0$ we

must recover the form factors of exponential fields [113]. Another consistency check for form factor solutions is provided by the fact that the conformal dimension $\Delta_n^\alpha$ of the field $\mathcal{T}_n^\alpha$ must be [49, 50, 115]

$$\Delta_n^\alpha = \frac{1}{24}\left(n - \frac{1}{n}\right) + \frac{\Delta_1^\alpha}{n}, \tag{61}$$

where $\Delta_1^\alpha = -\frac{g^2\alpha^2}{4(2\pi)^3}$ is the conformal dimension of the exponential field, which is negative in the shG model. As expected $\Delta_n^0 = \Delta_n$ as defined in (26).

Before presenting the two-particle FF solution $F_2^\alpha(\theta; B, n)$ that satisfies all constraints above, it is instructive to recall the BPTF solution $F_2^0(\theta; B, n)$, which appeared first in [33]

$$F_2^0(\theta; B, n) = \frac{\langle \mathcal{T}_n \rangle \sin \frac{\pi}{n}}{2n \sinh \frac{i\pi+\theta}{2n} \sinh \frac{i\pi-\theta}{2n}} \frac{R(\theta; B, n)}{R(i\pi; B, n)}, \tag{62}$$

where $\langle \mathcal{T}_n \rangle$ is the vacuum expectation value (VEV) of the BPTF and the minimal FF has the well-known formula [114]

$$R(\theta; B, n) = \exp\left[-2\int_0^\infty \frac{dt}{t} \frac{\sinh \frac{tB}{4} \sinh \frac{t(2-B)}{4}}{\sinh nt \cosh \frac{t}{2}} \cosh\left(t\left(n + \frac{i\theta}{\pi}\right)\right)\right], \tag{63}$$

and the property

$$\frac{R'(i\pi; B, 1)}{R(i\pi; \xi, 1)} = 2\int_0^\infty dt \frac{\cosh t \sinh \frac{tB}{4} \sinh \frac{t(2-B)}{4}}{\sinh^2 t \cosh \frac{t}{2}}, \tag{64}$$

where the prime means derivative w.r.t. $n$ evaluated at $n = 1$. Various representations of this function have been discussed in many papers, including our Part I (see subsection 5.1 and Appendix A), so we will not repeat them here. For the exponential field, the form factors are also known [113, 114] and take the simple form

$$F_2^\alpha(\theta; B, 1) = \langle e^{\frac{\alpha g \varphi}{2\pi}} \rangle \frac{4 \sin^2 \frac{\alpha B}{4}}{\sin \frac{\pi B}{2}} \frac{R(\theta; B, 1)}{R(i\pi; \xi, 1)}. \tag{65}$$

The two-particle FF of the exponential CTF turns out to have the expected structure, namely

$$F_2^\alpha(\theta; B, n) = F_2^0(\theta; B, n) + A(\alpha, n, B) \frac{R(\theta; B, n)}{R(i\pi; B, n)}, \tag{66}$$

where $A(\alpha, n, B)$ is a constant, that is, independent of $\theta$. This structure is easily justified. It is in fact the most general solution to the FF equations that possesses all desired properties. The first part obviously solves the equations, as it is the solution (62) whereas the second term is a minimal solution of the FF equations, since it is proportional to the minimal form factor. This additional term has no poles in the physical sheet, hence trivially satisfies the kinematic residue equation. Such a structure for the general solution of two-particle FF equations has been discussed for other local fields, including in [48] and in [53] for another composite field. Since

$$\lim_{\theta \to \infty} F_2^0(\theta; B, n) = 0 \quad \text{and} \quad \lim_{\theta \to \infty} R(\theta; B, n) = 1, \tag{67}$$

we have that the addition of the second term in (66) fundamentally changes the asymptotic properties of $F_2^\alpha(\theta; B, n)$ compared to $F_2^0(\theta; B, n)$. This change is important an in fact desirable because we expect

$$\lim_{\theta \to \infty} F_2^\alpha(\theta; B, n) = \frac{(F_1^\alpha(B, n))^2}{\langle \mathcal{T}_n^\alpha \rangle}, \tag{68}$$

where $F_1^\alpha(B, n)$ is the one-particle form factor which is rapidity-independent. This is a consequence of the clustering decomposition property of form factors which is discussed in more generality in [103]. This is consistent with the fact that, by symmetry considerations, the one-particle form factor of the exponential CTF is non-zero (whereas it is so for $\alpha = 0$). Indeed this form factor will become, under analytic continuation in $g$, the one-particle form factor of the lightest breather in the sG model.

We can now proceed to fixing the constant $A(\alpha, n, B)$ with the help of the following conditions

$$A(0, n, B) = 0, \quad A(\alpha, 1, B) = \langle e^{\frac{\alpha g \varphi}{2\pi}} \rangle \frac{4 \sin^2 \frac{\alpha B}{4}}{\sin \frac{\pi B}{2}}, \tag{69}$$

which are consequences of (62) and (65). We also know that the two-particle form factor above must solve a dynamical pole axion analogous to (51). A solution to these constraints is given by

$$A(\alpha, n, B) = \langle \mathcal{T}_n^\alpha \rangle \frac{2 \sin \frac{\pi}{2n} \sin^2 \frac{\alpha B}{4n}}{n \sin \frac{\pi B}{4n} \sin \frac{\pi(2-B)}{4n}}. \tag{70}$$

This solution can also be obtained using the method of angular quantisation (c.f. Appendix A). It then follows that the one-particle form factor of the exponential CTF is

$$F_1^\alpha(B, n) = \sqrt{\frac{\langle \mathcal{T}_n^\alpha \rangle A(\alpha, n, B)}{R(i\pi; B, n)}} = \langle \mathcal{T}_n^\alpha \rangle \sin \frac{\alpha B}{4n} \sqrt{\frac{2 \sin \frac{\pi}{2n}}{nR(i\pi; B, n) \sin \frac{\pi B}{4n} \sin \frac{\pi(2-B)}{4n}}}. \tag{71}$$

By construction $F_1^{\alpha \neq 0}(B, 1) \neq 0$ and there is a sign ambiguity in this solution (as we are taking a square root) but this does not affect any of the results in this paper, since all quantities of interest involve FFs squared.

## 4.1 Consistency Checks via $\Delta$-Sum Rule

The final solution (66) with (70) can be further checked by using the $\Delta$- sum rule [103] which states that the conformal dimension of a local field can be obtained from an integral involving its two-point function with the trace of the energy-momentum tensor $\Theta$. Let this local field be the exponential CTF in the shG theory. Then, expanding the two-point function in terms of form factors we have the general expression

$$\Delta_n^\alpha = -\frac{n}{2 \langle \mathcal{T}_n^\alpha \rangle} \sum_{k=1}^\infty \int_{-\infty}^\infty \frac{d\theta_1 \cdots d\theta_k}{(2\pi)^k k!} \frac{F_k^\Theta(\theta_1, \ldots, \theta_k; B) \left[ F_k^\alpha(\theta_1, \ldots, \theta_k; B, n) \right]^*}{\left( \sum_{p=1}^k m \cosh \theta_p \right)^2}, \tag{72}$$

where $m$ is the particle mass. In the shG model, the FFs of the stress-energy tensor $\Theta$ are non-vanishing only for even particle numbers and were computed in [114]. The two-particle FF is

$$F_2^\Theta(\theta; B) = 2\pi m^2 \frac{R(\theta; B, 1)}{R(i\pi; B, 1)}. \tag{73}$$

Thus, keeping only the two-particle contribution, which usually gives very accurate results [114], the formula above can be simplified to

$$\Delta_n^\alpha \approx -\frac{n}{32\pi^2 m^2 \langle \mathcal{T}_n^\alpha \rangle} \int_\infty^\infty d\theta \frac{F_2^\Theta(\theta; B) F_2^\alpha(\theta; B, n)^*}{\cosh^2 \frac{\theta}{2}}. \tag{74}$$

Table 1: The $\Delta$-sum rule in the two-particle approximation (SR) compared with the exact conformal dimension of the exponential CFT in the shG model (61) for various values of $\alpha$ and $B$. These include $\alpha = 0$, that is the standard BPTF and $n = 1$, $\alpha \neq 0$ corresponding to the exponential field in shG. In all cases the agreement is very good.

(a) $\alpha = 0.7039 \times 2\pi$, $B = 0.2$

| $n$ | $\Delta_n^0$ (Exact) | $\Delta_n^0$ (SR) | $\Delta_n^\alpha$ (Exact) | $\Delta_n^\alpha$ (SR) |
|---|---|---|---|---|
| 1 | 0 | 0 | -0.055053 | -0.053919 |
| 2 | 0.0625 | 0.063569 | 0.034974 | 0.034953 |
| 3 | 0.11111 | 0.113523 | 0.092760 | 0.094216 |
| 4 | 0.15625 | 0.159907 | 0.142487 | 0.145365 |
| 5 | 0.2 | 0.204842 | 0.188989 | 0.193184 |

(b) $\alpha = 0.4483 \times 2\pi$, $B = 0.4$

| $n$ | $\Delta_n^0$ (Exact) | $\Delta_n^0$ (SR) | $\Delta_n^\alpha$ (Exact) | $\Delta_n^\alpha$ (SR) |
|---|---|---|---|---|
| 1 | 0 | 0 | -0.050243 | -0.048283 |
| 2 | 0.0625 | 0.064081 | 0.037378 | 0.037318 |
| 3 | 0.11111 | 0.114828 | 0.094363 | 0.096610 |
| 4 | 0.15625 | 0.161949 | 0.143689 | 0.148183 |
| 5 | 0.2 | 0.207582 | 0.189951 | 0.196531 |

(c) $\alpha = -0.5623 \times 2\pi$, $B = 0.6$

| $n$ | $\Delta_n^0$ (Exact) | $\Delta_n^0$ (SR) | $\Delta_n^\alpha$ (Exact) | $\Delta_n^\alpha$ (SR) |
|---|---|---|---|---|
| 1 | 0 | 0 | -0.135506 | -0.120618 |
| 2 | 0.0625 | 0.064306 | -0.005253 | -0.007845 |
| 3 | 0.11111 | 0.115505 | 0.065942 | 0.065666 |
| 4 | 0.15625 | 0.163049 | 0.122373 | 0.125193 |
| 5 | 0.2 | 0.20908 | 0.172899 | 0.178615 |

We have evaluated this sum for various values of the replica index $n$, the interaction parameter $B$ and $\alpha$ and found very good agreement between the approximation (74) and (61), as demonstrated in Table 1. In this table (and others presented in Appendix B) , the exact and approximate sum rule (SR) dimensions of the exponential CTF are given. To judge the quality of the match between the exact and approximated values, it is worth comparing the data associated with the standard BPTF ($\alpha = 0$) with those associated with the exponential CTF for the same value of $B$. We see the same trends regarding the magnitude of the error: the difference is larger for larger $n$ and $B$ and the sum rule approximation consistently overshuts. The largest error percentage in any of the tables is of the order of %1. Altogether, the results of this section give strong evidence for the validity of the novel FF solutions. Additional tables with other parameter values are found in Appendix B.

## 5  Form factors of exponential CTFs in the sG theory

In this section we focus on the breather sector of the theory, where the $S$-matrices are diagonal. Our starting point are the $b_1$ form factors which are directly related to the FFs obtained in the previous section and in Appendix A. Denoting by

$$F_{b_1 \dots b_1}^\alpha (\theta_1, \dots, \theta_k; \xi, n), \tag{75}$$

the $k$-particle form factor associated to $k$ breathers of type $b_1$ of the $U(1)$ exponential CTF in the sG model, such form factor is related to the shG form factors by

$$F_{b_1 \dots b_1}^\alpha (\theta_1, \dots, \theta_k; \xi, n) := F_k^\alpha (\theta_1, \dots, \theta_k; B = -2\xi, n). \tag{76}$$

Recall that the exponential CTF in sG can be formally written in the usual way $\mathcal{T}_n^\alpha =: e^{\frac{i\alpha g \varphi}{2\pi}} \mathcal{T}_n :$, and we note again the presence of the $g$ parameter in the exponential. This factor ensures that $\alpha \in (-\pi, \pi]$ and that a soliton/antisoliton have $U(1)$ charge $\pm 1$.

As discussed in Part I [22] the $b_1$ FFs form the basis for the construction of heavier breather solutions thanks to the fusion procedure. Fusion is nothing but the repeated use of the bound state kinematic equation or dynamical pole axiom (51), given that each breather $b_k$ can be seen as a bound state of $k$ breathers $b_1$. Interpreting each arrow as an application of fusion, we have schematically

$$
\begin{aligned}
F^\alpha_{b_1 b_1 b_1 b_1}(\theta_1, \theta_2, \theta_3, \theta_4; \xi, n) &\mapsto F^\alpha_{b_2 b_1 b_1}(\theta_1, \theta_2, \theta_3; \xi, n) \\
&\mapsto F^\alpha_{b_2 b_2}(\theta; \xi, n) \quad \text{or} \quad F^\alpha_{b_3 b_1}(\theta; \xi, n) \mapsto F^\alpha_{b_4}(\xi, n), \quad (77)
\end{aligned}
$$

and

$$
F^\alpha_{b_1 b_1}(\theta; \xi, n) \mapsto F^\alpha_{b_2}(\xi, n). \tag{78}
$$

In [22] we argued that the above procedure (this is, step-by-step fusion) is equivalent to the prescription of [104,105] which describes the effect of simultaneously fusing multiple breathers. This can be expressed through the equation

$$
F^\alpha_{\underbrace{\ldots}_{p} b_k \underbrace{\ldots}_{r}}(\theta_1, \ldots, \theta_p, \theta, \theta_{p+2}, \ldots, \theta_{p+r+1}; \xi, n) = \left[ \prod_{i=1}^{k-1} \Gamma^{b_{i+1}}_{b_1 b_i} \right] \times \tag{79}
$$
$$
F^\alpha_{\underbrace{\ldots}_{p} \underbrace{b_1 \ldots b_1}_{k} \underbrace{\ldots}_{r}}(\theta_1, \ldots, \theta_p, \theta^{[k-1]}, \theta^{[k-3]}, \ldots, \theta^{[1-k]}, \theta_{p+2}, \ldots, \theta_{p+r+1}; \xi, n),
$$

where $\theta^{[a]} := \theta - \frac{i\pi\xi a}{2}$. In the following we will use the minimal form factor $R(\theta; \xi, n)$, which satisfies the equation

$$
R(\theta; \xi, n) = S_{b_1 b_1}(\theta) R(-\theta; \xi, n), \tag{80}
$$

and can be obtained (with some abuse of notation) from $R(\theta; B, n)$ by simply replacing $B = -2\xi$. The analytic properties of this function, notably the fact that, contrary to $R(\theta; B, n)$ it can have poles in the physical sheet, where discussed in much detail in our Part I [22].

## 5.1 One-particle form factors

The one-particle form factor $F^\alpha_{b_1}(\xi, n)$ can be obtained from (71) in the usual way. It is useful for us to write it in terms of a new constant

$$
F^\alpha_{b_1}(\xi, n) = -2\langle \mathcal{T}^\alpha_n \rangle \sin\frac{\alpha\xi}{2n} \mathcal{C}(\xi, n), \qquad \text{where} \qquad \mathcal{C}(\xi, n) = \sqrt{\frac{\sin\frac{\pi}{2n}}{2nR(i\pi; \xi, n)\sin\frac{\pi\xi}{2n}\sin\frac{\pi(1+\xi)}{2n}}}. \tag{81}
$$

Fig.1 shows some plots of this function, which takes real values. The one-particle $b_2$, $b_3$ and $b_4$ FFs can be obtained from higher particle $b_1$ FFs using the fusion technique [22]. In particular, we have

$$
\begin{aligned}
F^\alpha_{b_2}(\xi, n) &= \Gamma^{b_2}_{b_1 b_1} F^\alpha_{b_1 b_1}(-i\pi\xi; \xi, n) \\
&= \langle \mathcal{T}^\alpha_n \rangle \frac{\Gamma^{b_2}_{b_1 b_1} \sin\frac{\pi}{2n}}{n \sin\frac{\pi(1+\xi)}{2n}} \left[ \frac{\cos\frac{\pi}{2n}}{\sin\frac{\pi(\xi-1)}{2n}} - \frac{2\sin^2\frac{\xi\alpha}{2n}}{\sin\frac{\pi\xi}{2n}} \right] \frac{R(-i\pi\xi; \xi, n)}{R(i\pi; \xi, n)},
\end{aligned} \tag{82}
$$

$$
\begin{aligned}
F^\alpha_{b_3}(\xi, n) &= \Gamma^{b_2}_{b_1 b_1} \Gamma^{b_3}_{b_1 b_2} F^\alpha_{b_1 b_1 b_1}(-i\pi\xi, 0, i\pi\xi; \xi, n) = \langle \mathcal{T}^\alpha_n \rangle \Gamma^{b_2}_{b_1 b_1} \Gamma^{b_3}_{b_1 b_2} \mathcal{C}(\xi, n)^3 \\
&\quad \times 2 \left[ \sin\frac{3\alpha\xi}{2n} - \sin\frac{\alpha\xi}{2n} \frac{\sin\frac{\pi}{2n}\left(1 + 2\cos\frac{\pi\xi}{n}\right)}{\sin\frac{\pi(1-2\xi)}{2n}} \right] \times R(-i\pi\xi; \xi, n)^2 R(-i2\pi\xi; \xi, n),
\end{aligned} \tag{83}
$$

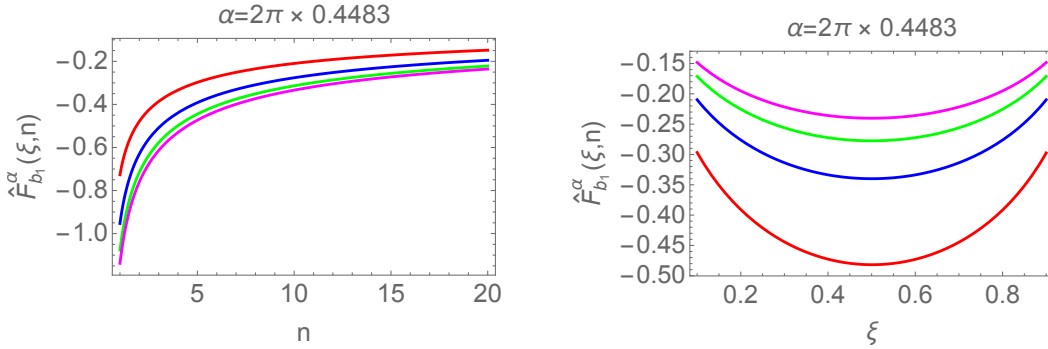

Figure 1: Left: The normalized one-particle form factor $\hat{F}^{\alpha}_{b_1}(\xi, n)$ as a function of $n$ for $\xi = 0.9$ (red), 0.8 (blue), 0.7 (green), 0.6 (magenta). Right: The normalized one-particle form factor $\hat{F}^{\alpha}_{b_1}(\xi, n)$ as a function of $\xi$ for $n = 5$ (red), 10 (blue), 15 (green), 20 (magenta). Normalized means that it has been divided by the VEV of the field.

and

$$
\begin{aligned}
F^{\alpha}_{b_4}(\xi, n) &= \Gamma^{b_2}_{b_1 b_1} \Gamma^{b_3}_{b_1 b_2} \Gamma^{b_4}_{b_1 b_3} F^{\alpha}_{b_1 b_1 b_1}\left(-\frac{i\pi 3\xi}{2}, -\frac{i\pi\xi}{2}, \frac{i\pi\xi}{2}, \frac{i\pi 3\xi}{2}; \xi, n\right) = \\
&= \langle \mathcal{T}^{\alpha}_n \rangle \mathcal{C}(\xi, n)^4 \Gamma^{b_2}_{b_1 b_1} \Gamma^{b_3}_{b_1 b_2} \Gamma^{b_4}_{b_1 b_3} \\
&\quad \times 2\left[\cos\frac{2\alpha\xi}{n} - 4\cos\frac{\pi\xi}{2n}\cos\frac{\pi\xi}{n}\cos\frac{\alpha\xi}{n}\frac{\sin\frac{\pi}{2n}}{\sin\frac{\pi(1-3\xi)}{2n}} + \right. \\
&\quad \left. +\frac{\sin\frac{\pi}{2n}\sin\frac{\pi(1-\xi)}{2n}\cos\frac{\pi\xi}{n}\left(1 + 2\cos\frac{\pi\xi}{n}\right)}{\sin\frac{\pi(1-2\xi)}{2n}\sin\frac{\pi(1-3\xi)}{2n}}\right] \\
&\quad \times R(-i\pi\xi; \xi, n)^3 R(-i2\pi\xi; \xi, n)^2 R(-i3\pi\xi; \xi, n).
\end{aligned}
\tag{84}
$$

It is easy to check that the one-particle FFs $F^0_{b_1}(\xi, n) = F^0_{b_3}(\xi, n) = 0$ and with some more algebra one can straightforwardly show that under the same limit $F^0_{b_2}(\xi, n)$ and $F^0_{b_4}(\xi, n)$ coincide with the BPTF solutions found in Part I [22]. It can be similarly checked that when $n = 1$, the FFs of the standard exponential field are recovered [12].

## 5.2 Two-Particle Form Factors

By analytic continuation from (66) with (70) we have

$$
F^{\alpha}_{b_1 b_1}(\theta; \xi, n) = \langle \mathcal{T}^{\alpha}_n \rangle \frac{2\sin\frac{\pi}{2n}}{n}\left[\frac{\cos\frac{\pi}{2n}}{2\sinh\frac{i\pi+\theta}{2n}\sinh\frac{i\pi-\theta}{2n}} + \frac{\sin\frac{\alpha x}{2n}}{\sin\frac{\pi\xi}{2n}\sin\frac{\pi(\xi+1)}{2n}}\right]\frac{R(\theta; \xi, n)}{R(i\pi; \xi, n)}.
\tag{85}
$$

The two-particle FF $F_{b_2 b_1}(\theta; \xi, n)$ can be computed by fusion as

$$
\begin{aligned}
F^{\alpha}_{b_2 b_1}(\theta; \xi, n) &= \Gamma^{b_2}_{b_1 b_1} F^{\alpha}_{b_1 b_1 b_1}\left(\theta - \frac{i\pi\xi}{2}, \theta + \frac{i\pi\xi}{2}, 0; \xi, n\right) = \langle \mathcal{T}^{\alpha}_n \rangle \Gamma^{b_2}_{b_1 b_1} \mathcal{C}(\xi, n)^3 \\
&\quad \times 2\left[\sin\frac{3\alpha\xi}{2n} + \text{Re}[P_{b_2 b_1}(\theta; \xi, n)]\right] \\
&\quad \times R\left(\theta - \frac{i\pi\xi}{2}; \xi, n\right)R\left(\theta + \frac{i\pi\xi}{2}; \xi, n\right)R(-i\pi\xi; \xi, n),
\end{aligned}
\tag{86}
$$

with

$$
P_{b_2 b_1}(\theta;\xi,n) :=
$$
$$
-i\frac{\sinh\frac{2\theta-3i\pi\xi}{4n}}{\sinh\frac{2\theta-i\pi(\xi-2)}{4n}}\left[e^{\frac{i\alpha\xi}{2n}}\frac{2\cos\frac{\pi\xi}{2n}\sin\frac{\pi}{2n}}{\sin\frac{\pi(\xi-1)}{2n}}\frac{\sinh\frac{2\theta+i\pi(\xi+2)}{4n}}{\sinh\frac{2\theta-i\pi\xi}{4n}}+e^{-\frac{i\alpha\xi}{2n}}\frac{\sinh\frac{2\theta+i\pi(3\xi+2)}{4n}}{\sinh\frac{2\theta+i\pi\xi}{4n}}\right]. \quad (87)
$$

Note that $\mathrm{Re}[P_{b_2 b_1}^\alpha(\theta;\xi,n)]$ is meant under the condition $\theta\in\mathbb{R}$. Finally we have also computed the two-particle form factor $F_{b_2 b_2}^\alpha(\theta;\xi,n)$

$$
\begin{aligned}
F_{b_2 b_2}^\alpha(\theta;\xi,n) &= \left(\Gamma_{b_1 b_1}^{b_2}\right)^2 F_{b_1 b_1 b_1 b_1}^\alpha\left(\theta-\frac{i\pi\xi}{2},\theta+\frac{i\pi\xi}{2},-\frac{i\pi\xi}{2},\frac{i\pi\xi}{2};\xi,n\right)\\
&= \langle\mathcal{T}_n^\alpha\rangle\left(\Gamma_{b_1 b_1}^{b_2}\right)^2 \mathcal{C}(\xi,n)^4 R(\theta-i\pi\xi;\xi,n)R(\theta+i\pi\xi;\xi,n)R(\theta;\xi,n)^2\\
&\quad \times R(-i\pi\xi;\xi,n)^2 \qquad\qquad\qquad\qquad\qquad\qquad\qquad\qquad (88)\\
&\quad \times\left[2\cos\frac{2\alpha\xi}{n}-4\cos\frac{\alpha\xi}{n}\frac{\sin\frac{\pi}{2n}\cos\frac{\pi\xi}{2n}\left(\cos\frac{\pi}{n}\cos\frac{\pi\xi}{n}-\sin^2\frac{\pi\xi}{n}-\cosh\frac{\theta}{n}\right)}{\sin\frac{\pi(\xi-1)}{2n}\sinh\frac{\theta+i\pi(\xi-1)}{2n}\sinh\frac{\theta-i\pi(\xi-1)}{2n}}\right.\\
&\quad \left.+2\mathrm{Re}\left[P_{b_2 b_2}(\theta;\xi,n)\right]+Y_{b_2 b_2}(\theta;\xi,n)\right],
\end{aligned}
$$

where

$$
P_{b_2 b_2}(\theta;\xi,n)=\frac{\sinh\frac{\theta-i\pi(\xi+1)}{2n}\sinh\frac{\theta-i\pi(2\xi+1)}{2n}\sinh\frac{\theta+i\pi\xi}{2n}\sinh\frac{\theta+2i\pi\xi}{2n}}{\sinh\frac{\theta}{2n}\sinh\frac{\theta-i\pi}{2n}\sinh\frac{\theta-i\pi\xi}{2n}\sinh\frac{\theta+i\pi(\xi-1)}{2n}}, \quad (89)
$$

and

$$
Y_{b_2 b_2}(\theta;\xi,n)=\left(\frac{\cos\frac{\pi\xi}{2n}\sin\frac{\pi}{2n}}{\sin\frac{\pi(\xi-1)}{4n}\cos\frac{\pi(\xi-1)}{4n}}\right)^2\frac{\sinh\frac{\theta+i\pi}{2n}\sinh\frac{\theta-i\pi}{2n}\sinh\frac{\theta+2i\pi\xi}{2n}\sinh\frac{\theta-2i\pi\xi}{2n}}{\sinh\frac{\theta+i\pi(\xi-1)}{2n}\sinh\frac{\theta-i\pi(\xi-1)}{2n}\sinh\frac{\theta+i\pi\xi}{2n}\sinh\frac{\theta-i\pi\xi}{2n}}. \quad (90)
$$

## 6 Symmetry Resolved Partition Functions and Entanglement

Having obtained the one- and two-particle form factor solutions in the breather sector we now have all we need to embark on our study of the symmetry resolved entanglement entropy. As we will see, the one-particle FF of the lightest breather $b_1$ provides the leading length-dependent correction to the SRE. For this reason and for its higher technical difficulty, we postpone a detailed study of the soliton-antisoliton sector to future work.

### 6.1 U(1) Charged Moments

In this subsection we discuss the $U(1)$ symmetry resolved charged moments in the sG theory. We restrict our analysis to a single-interval subsystem in the ground state of the full system. The charged moments as well as entropies can then be calculated from the two-point functions of the $U(1)$ CTFs. Specifying the subsystem as an interval $A=[0,\ell]$ the charged moments are written as

$$
Z_n(\alpha)=\mathrm{Tr}\left(\rho_A^n e^{i\alpha\hat{Q}_A}\right)=\varepsilon^{4\Delta_n^\alpha}\langle\mathcal{T}_n^\alpha(0)\tilde{\mathcal{T}}_n^\alpha(\ell)\rangle, \quad (91)
$$

in terms of an equal-time two-point function, where $\varepsilon$ is the UV regulator. In principle, there could be an additional $n$-dependent multiplicative constant, but this will play no role in later

sections, so we will not include it here. As seen earlier the conformal dimension $\Delta_n^\alpha$ is the same for $\mathcal{T}_n^\alpha$ and $\tilde{\mathcal{T}}_n^\alpha$: and is given by the same formula (61) up to analytic continuation $g \to ig$. The conformal dimensions are the sums of the scaling/conformal dimension of the standard BPTF and that of the $U(1)$ twist field divided by $n$ due to the effect of the branch point [49,50,52,75,76]. Note that the scaling dimension depends explicitly on the interaction parameter $g$ of the sG model. Eq. (61) also reproduces the known results for the free Dirac theory [74,75] at the free fermion point of the sG theory, that is when $g^2 = 4\pi$ or $\xi = 1$.

We now focus on the first few terms of the form factor expansion of the charged moments. In the attractive regime, those terms will be determined by the one-particle form factor of the lightest breather (that is, $b_1$ of mass $m_1$). We can write

$$Z_n(\alpha) = \varepsilon^{4\Delta_n^\alpha} \langle \mathcal{T}_n^\alpha(0) \tilde{\mathcal{T}}_n^\alpha(\ell) \rangle = (m\varepsilon)^{4\Delta_n^\alpha} D_n^\alpha \left( 1 + H_n^\alpha(m_1\ell) + \mathcal{O}(e^{-M\ell}) \right), \tag{92}$$

where

$$D_n^\alpha := m^{-4\Delta_n^\alpha} \langle \mathcal{T}_n^\alpha \rangle^2, \tag{93}$$

and $M$ is a mass scale which is either $M = m_2$ if the second breather is present or $M = 2m$ if it is not. This is so because it is easy to show that the mass of $b_2$ is, after $m_1$, the smallest mass scale that arises in the problem (this can be easily shown from the definition (12)).

Since the CTF has a non-vanishing one-particle FF contributions even when $n = 1$ we have that, instead of the standard two-particle approximation, here we can obtain the next-to-leading order behaviour in $\ell$ of the two-point function from the first breather one-particle FF. More precisely, a standard form factor expansion, gives

$$\langle \mathcal{T}_n^\alpha(0) \tilde{\mathcal{T}}_n^\alpha(\ell) \rangle \approx \langle \mathcal{T}_n^\alpha \rangle^2 + \sum_{j=1}^n \int_{-\infty}^\infty \frac{\mathrm{d}\theta}{(2\pi)} |F_{b_1}^\alpha(\xi, n)|^2 e^{-\ell m_1 \cosh\theta} + \mathcal{O}(e^{-M\ell})$$
$$= \langle \mathcal{T}_n^\alpha \rangle^2 \left( 1 + \frac{n}{\pi} |\hat{F}_{b_1}^\alpha(\xi, n)|^2 K_0(m_1\ell) + \mathcal{O}(e^{-M\ell}) \right), \tag{94}$$

where $K_0(x)$ is the modified Bessel function of the second kind and the "hatted" FF is the form factor normalized by $\langle \mathcal{T}_n^\alpha \rangle$. Therefore we can identify

$$H_n^\alpha(m_1\ell) = \frac{n}{\pi} |\hat{F}_{b_1}^\alpha(\xi, n)|^2 K_0(m_1\ell). \tag{95}$$

The $n$-derivative of this function evaluated at $n = 1$ plays a role in higher order corrections to the symmetry resolved entanglement entropy

$$[\partial_n H_n^\alpha(m_1\ell)]_{n=1} = \frac{1}{\pi} \left[ |\hat{F}_{b_1}^\alpha(\xi, 1)|^2 + [\partial_n |\hat{F}_{b_1}^\alpha(\xi, n)|^2]_{n=1} \right] K_0(m_1\ell), \tag{96}$$

and can be computed explicitly from the results of Subsection 5.1. Since the form factor takes real values, we can just compute

$$(\hat{F}_{b_1}^\alpha(\xi, 1))^2 = 4\sin^2 \frac{\alpha\xi}{2} \mathcal{C}(\xi, 1)^2 = \frac{4\sin^2 \frac{\alpha\xi}{2}}{R(i\pi; \xi, 1)\sin \pi\xi}, \tag{97}$$

and

$$[\partial_n(\hat{F}_{b_1}^\alpha(\xi, n))^2]_{n=1} = \tag{98}$$
$$-(\hat{F}_{b_1}^\alpha(\xi, 1))^2 \left[ 1 + \alpha\xi \cot \frac{\alpha\xi}{2} - \frac{\pi(1+2\xi)}{2} \cot \pi\xi + \frac{\pi}{2} \csc \pi\xi + \frac{R'(i\pi; \xi, 1)}{R(i\pi; \xi, 1)} \right],$$

where the derivative of $R'(i\pi; \xi, 1)$ is a real number given by (64) with $B = -2\xi$. Focusing still on the contribution of the first breather, a useful result in the next section will be the small $\alpha$ expansion

$$(\hat{F}_{b_1}^\alpha(\xi, n))^2 = \mathcal{F}_{b_1}^{(2)}(\xi, n)\alpha^2 + \mathcal{O}(\alpha^4), \quad \text{with} \quad \mathcal{F}_{b_1}^{(2)}(\xi, n) = \frac{\xi^2 \sin\frac{\pi}{2n}}{n^3 R(i\pi; \xi, n)\sin\frac{\pi\xi}{2n}\sin\frac{\pi(1+\xi)}{2n}}. \tag{99}$$

In summary, from (92) and (94) and noting that

$$(m\varepsilon)^{4\Delta_1^\alpha} D_1^\alpha = \varepsilon^{4\Delta_1^\alpha}\langle e^{\frac{ig\alpha\varphi}{2\pi}}\rangle^2, \tag{100}$$

with $\Delta_1^\alpha = \frac{g^2\alpha^2}{32\pi^3}$ the scaling dimension of the sG exponential field and

$$\partial_n\left[(m\varepsilon)^{4\Delta_n^\alpha} D_n^\alpha\right]_{n=1} = \varepsilon^{4\Delta_1^\alpha}\langle e^{\frac{ig\alpha\varphi}{2\pi}}\rangle^2\left[\frac{\hat{c}}{3}\ln(m\varepsilon) + \frac{2[\partial_n\langle\mathcal{T}_n^\alpha\rangle]_{n=1}}{\langle e^{\frac{ig\alpha\varphi}{2\pi}}\rangle}\right], \tag{101}$$

where $\hat{c} := 1 - 12\Delta_1^\alpha$ so we can write

$$\begin{aligned}[\partial_n Z_n(\alpha)]_{n=1} &= \varepsilon^{4\Delta_1^\alpha}\langle e^{\frac{ig\alpha\varphi}{2\pi}}\rangle^2\left[\left[\frac{\hat{c}}{3}\ln(m\varepsilon) + \frac{2[\partial_n\langle\mathcal{T}_n^\alpha\rangle]_{n=1}}{\langle e^{\frac{ig\alpha\varphi}{2\pi}}\rangle}\right]\left[1 + H_1^\alpha(m_1\ell)\right]\right.\\ &\quad \left. + [\partial_n H_n^\alpha(m_1\ell)]_{n=1} + \mathcal{O}(e^{-M\ell})\right].\end{aligned} \tag{102}$$

We end this section by recalling some special limits of the formulae above, corresponding to the standard BPTF. First, we note that (up to a constant, as mentioned at the beginning of the section) $(1-n)^{-1}\ln(Z_n(0))$ is nothing but the standard Rényi entropy and, expanding the logarithm, this can be written as

$$S_n(\ell) = -\frac{1+n}{6n}\ln(m\varepsilon) + \frac{\ln D_n^0}{1-n} + \frac{1}{1-n}H_n^0(m_1\ell) + \mathcal{O}(e^{-M\ell}). \tag{103}$$

One of the main results of [33] was the realization that the limit $n \to 1$ of the function above is non-trivial in the sense that in that limit one-particle form factor contributions are all vanishing and the leading large-distance correction comes from two-particle form factor contributions and takes a universal form

$$S(\ell) = -\frac{1}{3}\ln(m\varepsilon) + U - \frac{y}{8}K_0(2\tilde{M}\ell) + \mathcal{O}(e^{-2\hat{M}\ell}), \tag{104}$$

where $U$ is the universal constant

$$U := [\partial_n(1-n)^{-1}\ln D_n^0]_{n=1}, \tag{105}$$

and, $y$, $\tilde{M}$ and $\hat{M}$ depend on the relative value of the masses of the soliton and the first breather. More precisely, $y = 2$ and $\tilde{M} = m$ if $m < m_1$, in which case $\hat{M} = m_1$. However, for small enough $\xi$ we can also have $m_1 < m$ in which case the leading correction has $y = 1$, $\tilde{M} = m_1$ and the first subleading correction will involve $\hat{M} = m$. The leading contribution was numerically confirmed in [51] in the repulsive regime where it corresponds to the solition/antisoliton.

## 6.2 U(1) Symmetry Resolved Entanglement

To turn to symmetry resolved entropies, let us start by recalling the definition of the symmetry resolved partition functions (8) in terms the charged moments (7):

$$\mathcal{Z}_n(q) = \int_{-\pi}^{\pi}\frac{d\alpha}{2\pi}Z_n(\alpha)e^{-i\alpha q}. \tag{106}$$

In order to perform the Fourier transform of equation (102), we generally need to know $\langle \mathcal{T}_n^{\alpha} \rangle$, which we do not know for the sG model. Luckily, the precise form of this quantity can be ignored in many important cases, e.g. when one is interested in the difference between symmetry resolved and conventional entropies. This will be the focus of this section.

As we see from the (91) the integrand of (106) is proportional to $(m\varepsilon)^{4\Delta_n^{\alpha}} = e^{4\Delta_n^{\alpha} \ln(m\varepsilon)}$. As seen earlier, $\Delta_n^{\alpha}$ depends quadratically on the integration variable $\alpha$. At the same time we are interested in the regime where $m\varepsilon$ is a small quantity, as $\varepsilon$ is a small UV cut-off. This means that the leading contribution to (106) can be obtained from a saddle-point analysis. In other words, the main contribution to the integral will come from values of $\alpha$ near zero. This also means that in order to obtain the leading contribution to the integral it is sufficient to expand the other factors in $Z_n(\alpha)$ around $\alpha = 0$. We have for instance that

$$D_n^{\alpha} = D_n^0 + \alpha^2 D_n^{(2)} + \mathcal{O}(\alpha^4), \tag{107}$$

with $D_n$ as defined after (103). The absence of an $\mathcal{O}(\alpha)$ term is justified on symmetry grounds. Combining this with (99) we can write

$$\mathcal{Z}_n(q) \approx (m\varepsilon)^{4\Delta_n^0} \int_{-\pi}^{\pi} \frac{d\alpha}{2\pi} (m\varepsilon)^{\frac{\Delta\alpha^2}{n}} \left[ D_n^0 + \alpha^2 \left[ D_n^{(2)} + \frac{\pi D_n^0}{n} \mathcal{F}_{b_1}^{(2)}(\xi, n) K_0(m_1 \ell) \right] + \mathcal{O}(\alpha^4) \right] e^{-i\alpha q}, \tag{108}$$

where $\Delta = \frac{g^2}{4(2\pi)^3}$. Integrating and then expanding around $m\varepsilon = 0$ we obtain

$$
\begin{aligned}
\mathcal{Z}_n(q) = (m\varepsilon)^{4\Delta_n^0} \Bigg[ & \frac{D_n^0 \sqrt{n} \, e^{\frac{-nq^2}{4\Delta |\ln(m\varepsilon)|}}}{2\sqrt{\pi\Delta}\sqrt{|\ln(m\varepsilon)|}} \\
& + \left[ D_n^{(2)} + \frac{\pi D_n^0}{n} \mathcal{F}_{b_1}^{(2)}(\xi, n) K_0(m_1 \ell) \right] \frac{n^{3/2} e^{-\frac{nq^2}{4\Delta |\log(m\epsilon)|}} (nq^2 - 2\Delta |\ln m\epsilon|)}{8\sqrt{\pi}\Delta^{5/2} |\ln(m\epsilon)|^{5/2}} \\
& + \mathcal{O}\left( |\ln(m\varepsilon)|^{-\frac{5}{2}}, q^2 |\ln(m\varepsilon)|^{-\frac{7}{2}}, (m\epsilon)^{\frac{\pi^2\Delta}{n}} |\ln(m\epsilon)|^{-1}, e^{-M\ell} \right) \Bigg],
\end{aligned} \tag{109}
$$

making the assumption that $|\ln(m\varepsilon)| \gg (m\epsilon)^{\frac{\pi^2\Delta}{n}} q^2$, which allows us to simplify our expressions by taking the limiting values of some erf functions or equivalently, to extend the range of integration from $[-\pi, \pi]$ to $(-\infty, \infty)$. The $\mathcal{O}(q^2 |\ln(m\varepsilon)|^{-\frac{7}{2}})$ term originates from the neglected $\alpha^4$ part in (107) and by demanding that $q^2 \ll |\ln(m\varepsilon)|$ it can be safely omitted. This way we arive at

$$
\begin{aligned}
\mathcal{Z}_n(q) = (m\varepsilon)^{4\Delta_n^0} e^{\frac{-nq^2}{4\Delta |\ln(m\varepsilon)|}} \Bigg[ & \frac{D_n^0 \sqrt{n}}{2\sqrt{\pi\Delta}\sqrt{|\ln(m\varepsilon)|}} \\
& + \left[ D_n^{(2)} + \frac{\pi D_n^0}{n} \mathcal{F}_{b_1}^{(2)}(\xi, n) K_0(m_1 \ell) \right] \frac{n^{3/2} (nq^2 - 2\Delta |\ln m\epsilon|)}{8\sqrt{\pi}\Delta^{5/2} |\ln(m\varepsilon)|^{5/2}} \\
& + \mathcal{O}\left( |\ln(m\varepsilon)|^{-\frac{5}{2}}, (m\epsilon)^{\frac{\pi^2\Delta}{n}} |\ln(m\varepsilon)|^{-1}, e^{-M\ell} \right) \Bigg],
\end{aligned} \tag{110}
$$

where we kept only the leading $q$-dependence besides the Gaussian factor. From either (109) or (110) we can then work out an expression for the symmetry resolved Rényi entropies (5), that is

$$S_n(q, \ell) = -\frac{n+1}{6n} \ln(m\varepsilon) + \frac{\ln \sqrt{n} D_n^0}{1-n} - \frac{1}{2} \ln \frac{\Delta |\ln(m\varepsilon)|}{\pi} + \mathcal{O}(|\ln(m\varepsilon)|^{-\frac{1}{2}}, e^{-m_1 \ell}). \tag{111}$$

If we subtract from this, the standard Rényi entropy (103) we find that the leading terms are cancelled out and we obtain

$$S_n(q, \ell) - S_n(\ell) = \frac{1}{2} \frac{\ln n}{1-n} - \frac{1}{2} \ln \frac{\Delta |\ln(m\varepsilon)|}{\pi} + \mathcal{O}(|\ln(m\varepsilon)|^{-\frac{1}{2}}, e^{-m_1 \ell}). \tag{112}$$

One can easily determine from (109) also the leading term with $q$-dependence, which reads for $S_n(q)$ as

$$S_n(q,\ell) - S_n(q=0,\ell) = \frac{1}{1-n}\left[\frac{D_n^{(2)}}{D_n^0}n^2 - \frac{D_1^{(2)}}{D_1^0}n^3\right]\frac{q^2}{4\Delta^2|\ln(m\varepsilon)|^2} + \mathcal{O}(|\ln(m\varepsilon)|^{-3}), \quad (113)$$

and we remind ourselves that the above formula is valid when $|\ln(m\varepsilon)| \gg q^2$ and $\ell$ is large. From these expressions we can also obtain the corresponding von Neumann entropies, that is

$$S(q,\ell) = -\frac{1}{3}\ln(m\varepsilon) - \frac{1}{2} + U - \frac{1}{2}\ln\frac{\Delta|\ln(m\varepsilon)|}{\pi} + \mathcal{O}(|\ln(m\varepsilon)|^{-\frac{1}{2}}, e^{-2m\ell}), \quad (114)$$

where the order $e^{-2m\ell}$ is justified for the same reasons as in the standard entanglement entropy. The difference between the symmetry resolved and standard von Neumann entropies is

$$S(q,\ell) - S(\ell) = -\frac{1}{2} - \frac{1}{2}\ln\frac{\Delta|\ln(m\varepsilon)|}{\pi} + \mathcal{O}(|\ln(m\varepsilon)|^{-\frac{1}{2}}, e^{-2\tilde{M}\ell}). \quad (115)$$

At this order, we observe equipartition of entanglement, namely that $S_n(q,\ell)$ and $S(q,\ell)$ do not depend on $q$, in other words the symmetry resolved entanglement is equally distributed among symmetry sectors of the theory. The equipartition is, nevertheless, broken explicitly for finite $\varepsilon$ with the largest subleading term

$$S(q,\ell) - S(q=0,\ell) = -\left[\left(\frac{D_n^{(2)}}{D_n^0}\right)' + \frac{D_1^{(2)}}{D_1^0}\right]\frac{q^2}{4\Delta^2|\ln(m\varepsilon)|^2} + \mathcal{O}(|\ln(m\varepsilon)|^{-3}), \quad (116)$$

which is suppressed in $|\ln(m\varepsilon)|$ as $|\ln(m\varepsilon)|^{-2}$ when $|\ln(m\varepsilon)| \gg q^2$ and $\ell$ is large. Our results (112) and (115) are consistent with the findings of [74]. In particular, Eqs. (112) and (115) reproduce the expressions for the Dirac field theory at the free fermion point. This corresponds to setting $g^2 = 4\pi$. It is important to note that when $\xi$ approaches zero, so does $g^2$, which results in a logarithmic singularity at $\xi = 0$ in the difference of symmetry resolved and conventional entropies. This limit, nevertheless, is not physically sensible since it corresponds to the real non-compactified massive free boson theory, for which $U(1)$ symmetry is no longer present.

Finally we mention that the total von Neumann entropy can be written as the sum of the configurational and fluctuation (or number) entropy as [119]

$$S = \sum_{q_A} p(q_A)S(q_A) - \sum_{q_A} p(q_A)\ln p(q_A) = S^c + S^f, \quad (117)$$

as well, where $p(q_A) = \mathcal{Z}_1(q_A)$ equals the probability of finding $q_A$ as the outcome of a measurement of the symmetry operator restricted to a subsystem $\hat{Q}_A$. The contribution $S^c$ denotes the configurational entanglement entropy and measures the total entropy due to each charge sector weighted with the corresponding probability [64,120]. $S^f$ denotes the fluctuation entanglement entropy and is associated with the entropy due to the fluctuations of the value of the charge in the subsystem $A$ [64,90,121,123]. One can notice that in Eqs.(112) and (115) the $\ln\ln$ term is actually necessary in the symmetry resolved entropy in order to cancel the same contribution to the total entropy originating from $S_f$.

## 7  Conclusions

In this paper we have extended our study of entanglement measures in the sine-Gordon model to the symmetry resolved entanglement entropy. It has been known for some time [63] that the

symmetry resolved entropies can be expressed in terms of correlation functions of composite branch point twist fields. Composition here is meant in the sense first described in [49,50,53], namely that a composite twist field can be defined as the leading field in the operator product expansion of a branch point twist field and another primary field in CFT. The composite twist fields considered in this paper can been seen as massive extensions of such operators. In particular, when branch point twist fields are composed with another field which is also a twist field, the resulting form factors satisfy a new set of equations, first formulated in [76]. Here, for the first time we extend that formulation to non-diagonal theories and adapt it to the $U(1)$ twist field which implements the relevant internal symmetry of the sine-Gordon model.

Starting from these equations we find their solutions for the breather sector of the theory. This sector is neutral with respect to the $U(1)$ symmetry so that the form factor equations reduce to those of the standard branch point twist field and we can solve them for instance by using the angular quantisation method, as reviewed in Appendix A. Even if the equations are the same, we still find new solutions compared to those presented in Part I. This is because the asymptotic properties of the form factors of composite twist fields are necessarily different. This holds even for the two-particle form factors, where there are now additional terms that ensure these are non-vanishing in the limit of infinite rapidity. This non-vanishing limit is equivalent to non-vanishing one-particle form factors, which we compute for the first four breathers. We test our solutions against the $\Delta$-sum rule, finding very good agreement within the two-particle approximation. In this part of the work we use extensively the correspondence between the sinh-Gordon and sine-Gordon models under analytic continuation of the coupling constant. This allows us to use form factor solutions in the sinh-Gordon model as a basis for the construction of solutions in the sine-Gordon theory.

Finally, we employ these solutions to study the symmetry resolved entanglement entropies. We compute the corresponding charged momenta in terms of the two-point function of composite twist fields and use a saddle point approximation around charge zero to evaluate their transform, that is the corresponding partition function. For the charged momenta we show that for large region size they saturate to a mass/gap dependent function and that size-dependent corrections to this saturation constant are dominated by the one-particle form factor contributions in the breather sector, with the first breather providing the leading exponentially decaying contribution. This holds for replica index $n \in \mathbb{R}$ and $n \geq 1$ since the one-particle form factors are analytic functions of $n$ and therefore the analytic continuation to $n = 1$ is trivial for these terms. As mentioned above, the quantities we consider here involve two-point functions only and therefore our results are expected to hold for the Thirring model as well.

An important open problem remaining from this study is the computation of form factors of the CTF in the soliton/antisoliton sector. In this paper we have not addressed this problem for two main reasons: first, because form factor contributions to the symmetry resolved entropies from the soliton/antisoliton sector will always be subleading compared to those from the breather sector, thus they are not essential for the main application we considered in this paper and, second, because the study of the soliton/antisoliton sector is technically much more challenging. Concerning their more challenging nature, this is due to the fact that they satisfy truly distinct equations where the $U(1)$ phases are involved. This seemingly small change in the monodromy properties of the form factors, introduces a major challenge when it comes to solving the FF equations. Examining the FF solutions given in [12] for the exponential field gives a clue as to the difficulty. The FFs do no longer depend on simple integer powers of the variables $e^{\theta}$ (or $e^{\frac{\theta}{n}}$ in the replica model) but rather on powers involving the $U(1)$ charge and the sG coupling $\xi$. This expands the space of possible functions, making the functional form of the FFs much harder to constrain and renders standard FF solution methods based on simple ansatzs quite inadequate. In fact, it is in this context that the power of the angular quantisa-

tion method becomes apparent, as it allows for determining the FFs unambiguously. We hope to return to this problem in the future and to extend the angular quantisation approach to the exponential CTFs.

### Acknowledgments

We are grateful to Benjamin Doyon for useful discussions. PC and DXH acknowledge support from ERC under Consolidator grant number 771536 (NEMO).

## A  Angular quantisation scheme for the sinh-Gordon model

In this appendix we present a compact and straightforward derivation of the FFs $F_k^\alpha(\underline{\theta}; B, n)$ of the exponential composite branch point twist field for any particle number. The present derivation is based on the free field representation technique [11], which was first applied for the shG model in [122] and was generalised to obtain the FFs of conventional BPTFs of the shG model in [33]. We review this construction only for the specific case of the shG model in a similar fashion to that of appendix A in [33].

In the angular quantisation scheme, one quantise the model on radial half-lines starting from some fixed point in space. Accordingly the Hilbert space in this quantisation is understood as a subspace of the space of field configurations on those radial half-lines and the Hamiltonian (usually associated with time translations) generates rotations in this scheme. The main advantage and the power of this method is that for many integrable models, the angular Hilbert space has a simple structure, in particular it is a Fock-space $\mathcal{F}$ generated by bosonic oscillator modes. Additionally, for many models an explicit embedding is known for the states living in the conventional Hilbert space $\mathcal{H}$ in the usual quantisation scheme into the space of operators acting on the Fock-space $\mathcal{F}$. The correspondence between quantum states of $\mathcal{H}$ and operators of End($\mathcal{F}$) is particularly useful to compute FFs as we demonstrate below.

Specifying our treatment to the shG model, we consider the free oscillator modes $\lambda_\nu$ satisfying the following commutation relations

$$[\lambda_\nu, \lambda_{\nu'}] = \delta(\nu + \nu')f(\nu), \quad \text{with} \quad f(\nu) = \frac{2 \sinh\frac{\pi B \nu}{4} \sinh\frac{\pi(2-B)\nu}{4}}{\nu \cosh\frac{\pi \nu}{2}}. \tag{118}$$

These oscillator modes can be regarded as the modes of a free field. Additionally, therefore, we have to encounter the zero modes of the field $P$ and $Q$ as well, which satisfy the usual relation

$$[P, Q] = -i. \tag{119}$$

The Hamiltonian of the system, which we denote by $K$, can be written in terms of the oscillator modes as

$$K = \int_0^\infty \mathrm{d}\nu \frac{\nu}{f(\nu)} \lambda_{-\nu} \lambda_\nu, \tag{120}$$

and hence

$$[K, \lambda_\nu] = -\nu \lambda_\nu, \tag{121}$$

as in free theories. The Fock-space, i.e., the angular Hilbert space can be naturally written as

$$\mathcal{F} = \bigoplus_p \mathcal{F}_p, \tag{122}$$

where $p$ is the eigenvalue of the zero mode $P$ and $\mathcal{F}_p$ is spanned by the creation operators.

To elaborate on the embedding of $\mathcal{H}$ into $\text{End}(\mathcal{F})$ first we note that vacuum expectation values of operators $O$ in $\mathcal{H}$ can be identified with traces on $\mathcal{F}$, as a consequence of changing quantisation scheme [122]. Consequently we can write

$$\langle 0|O|0\rangle = \frac{\text{Tr}\left(e^{-2\pi K}\omega(O)\right)}{\text{Tr}\left(e^{-2\pi K}\right)} =: \langle\langle\omega(O)\rangle\rangle, \tag{123}$$

where $O$ is a product of local fields on the representation on $\mathcal{H}$ and $\omega(O)$ is composed of the same field represented on $\mathcal{F}$. The embedding is reflected by the Zamolodchikov-Faddeev (ZF) operators $Z(\theta)$, which now act on the angular Hilbert space $\mathcal{F}$ and their products correspond to the usual asymptotic states of $\mathcal{H}$. The operators $Z(\theta)$ are defined as follows

$$Z(\theta) = -iC\left[e^{i\pi P/\mathcal{Q}}\Lambda^+(\theta + i\pi/2) - e^{-i\pi P/\mathcal{Q}}\Lambda^-(\theta - i\pi/2)\right], \tag{124}$$

with

$$\mathcal{Q} = B/(2g), \tag{125}$$

and

$$\Lambda^\eta(\theta) =: e^{-i\eta\int d\nu\lambda_\nu e^{i(\theta-i\pi/2)}}:, \tag{126}$$

which ensure the exchange relations of the ZF algebra as well. This way we have obtained a representation of the ZF algebra in terms of free bosonic field.

Concerning the representation $\omega(O)$ on $\mathcal{F}$, it is expected that any field $O$ at the origin that is local with respect to the fundamental field, $\omega(O)$ commutes with the ZF operators. In particular, the FFs of the exponential fields $e^{g\alpha\varphi/(2\pi)}$ at the origin are obtained by choosing $\omega(e^{g\alpha\varphi/(2\pi)})$ to be a projector on $P$ with eigenvalue $p = g\alpha/(2\pi)$, and the projector is multiplied by the VEV of the exponential field as well [122]. Before demonstrating how the FFs of exponential fields can be obtained, we eventually need to compute quantities like $\langle\langle Z(\theta_1)Z(\theta_2)\rangle\rangle$. Evaluating first the trace for the oscillator modes, we find that

$$\langle\langle\lambda_\nu\lambda_{\nu'}\rangle\rangle = \frac{f(\nu)}{1-e^{-2\pi\nu}}\delta(\nu+\nu'), \tag{127}$$

from which, together with the expectation value of exponentials of free fields

$$\langle\langle : e^{\int d\nu\lambda_\nu a(\nu)} :: e^{\int d\nu\lambda_\nu b(\nu)} :\rangle\rangle = \exp\left(\int d\nu d\nu' a(\nu)b(\nu')\langle\langle\lambda_\nu\lambda_{\nu'}\rangle\rangle\right), \tag{128}$$

it follows that

$$\langle\langle\Lambda^{\eta_1}(\theta_1 + i\eta_1\pi/2)\Lambda^{\eta_2}(\theta_2 + i\eta_2\pi/2)\rangle\rangle =$$
$$\exp\left[-2\eta_1\eta_2\int_0^\infty \frac{dt}{t}\frac{\sinh\frac{\pi B}{4}\sinh\frac{\pi(2-B)}{4}}{\sinh t\cosh\frac{t}{2}}\cosh\left(t\left(1+i\frac{\theta_1-\theta_2}{\pi}-\frac{1}{2}(\eta_1-\eta_2)\right)\right)\right]$$
$$= \begin{cases} R(\theta_1-\theta_2;B,1) & \text{if } \eta_1 = \eta_2 \\ \Phi(\theta_1-\theta_2;B,1)R(\theta_1-\theta_2;B,1) & \text{if } \eta_1 = -\eta_2 = 1 \\ \Phi(2\pi i-\theta_1+\theta_2;B,1)R(\theta_1-\theta_2;B,1) & \text{if } \eta_1 = -\eta_2 = -1 \end{cases}, \tag{129}$$

with

$$\Phi(\theta;B,n) = -\frac{\cos\frac{\pi(B-1)}{2n} - \cos\frac{\pi-2i\theta}{2n}}{2i\sin\frac{\pi-i\theta}{2n}\sinh\frac{\theta}{2n}}. \tag{130}$$

Denoting the projector associated with the $p = g\alpha/(2\pi)$ eigenvalue of $P$ by $\mathcal{P}(g\alpha/(2\pi))$ we can define

$$
\begin{aligned}
Z_\alpha(\theta) &= \mathcal{P}(g\alpha/(2\pi))Z(\theta)\mathcal{P}(g\alpha/(2\pi)), \\
Z_\alpha(\theta) &= -iC\left[e^{i\pi\alpha B/(4\pi)}\Lambda^+(\theta + i\pi/2) - e^{-i\pi\alpha B/(4\pi)}\Lambda^-(\theta - i\pi/2)\right],
\end{aligned}
\tag{131}
$$

and the FFs of the exponential field $e^{g\alpha\varphi/(2\pi)}$ can now be expressed in terms of (129) and Wick's theorem as

$$
\begin{aligned}
\langle 0 \mid e^{\frac{\alpha}{2\pi}g\varphi} \mid \theta_1, \dots \theta_k\rangle &= \langle\langle Z_\alpha(\theta_1) \dots Z_\alpha(\theta_k)\rangle\rangle \\
&= \langle e^{\frac{\alpha}{2\pi}g\varphi}\rangle C^k \sum_{\eta_1,\dots,\eta_k} \exp\left[(i\pi\alpha B/(4\pi) - i\pi/2)\sum_{l=1}^{k}\eta_l\right] \\
&\quad \prod_{i<j}\langle\langle\Lambda^{\eta_i}(\theta_i + i\eta_i\pi/2)\Lambda^{\eta_j}(\theta_j + i\eta_j\pi/2)\rangle\rangle.
\end{aligned}
\tag{132}
$$

The FFs of the standard BPTF can be obtained via angular quantisation as well [33]. It was noticed in [33] that for twist fields associated with a symmetry one can write

$$
\omega(\mathcal{T}_\sigma(0)) = \langle\mathcal{T}_\sigma\rangle\sigma_\mathcal{F},
\tag{133}
$$

where $\sigma_\mathcal{F}$ is the action of the symmetry on the Fock-space, since the equal-time slices of angular quantisation are just the half lines originating from $(0,0)$. To turn the case of BPTFs let us now consider the n-copy sinh-Gordon model. We have a new angular quantisation Hilbert space

$$
\bigoplus_j \mathcal{F}^{(j)},
\tag{134}
$$

where $\mathcal{F}^{(j)}$ corresponding to the different replicas are isomorphic to $\mathcal{F}$. Accordingly we have different ZF operators $Z_j(\theta)$, for $j = 1, 2, \dots, n$, which are made up the bosonic modes $\lambda_{j,\nu}$, with now $j = 1, 2, \dots, n$. These operators commute for different values of $j$. Specifying the symmetry as $\sigma : j \leftrightarrow j+1 \mod n$, we can write for generic FFs

$$
\langle 0|\mathcal{T}_n(0,0)|\theta_1, \dots \theta_k\rangle_{\mu_1 \dots \mu_k} = \langle\mathcal{T}_n\rangle C_\sigma^k\langle\langle Z_{\mu_1}(\theta_1) \dots Z_{\mu_k}(\theta_k)\rangle\rangle_\sigma,
\tag{135}
$$

with

$$
\langle\langle\bullet\rangle\rangle_\sigma = \frac{\langle\langle\sigma_\mathcal{F}\bullet\rangle\rangle}{\langle\langle\sigma_\mathcal{F}\rangle\rangle}.
\tag{136}
$$

In the above formula, we have introduced a new constant $C_\sigma^k$, because we implicitly re-defined normal ordering to ensure that the computation of the trace of $Z$ products goes as before. This change of normal-ordering merely changes the normalisation, and the operators $\Lambda_\mu^\pm(\theta)$ in a uniform manner. As shown in Ref. [33] $\sigma_\mathcal{F}\lambda_{j,\nu}\sigma_\mathcal{F}^{-1} = \lambda_{j+1,\nu}$, and due to the cyclic properties of the trace one finds that

$$
\langle\langle\lambda_{j,\nu}\lambda_{1,\nu'}\rangle\rangle_\sigma = \frac{e^{-2\pi\nu(j-1)}f(\nu)}{1 - e^{-2\pi n\nu}}\delta(\nu + \nu').
\tag{137}
$$

As for the BPTF, that the eigenvalue of $P$ is zero we have due to (137) that

$$\langle\langle\Lambda_1^{\eta_1}(\theta_1 + i\eta_1\pi/2)\Lambda_1^{\eta_2}(\theta_2 + i\eta_2\pi/2)\rangle\rangle_\sigma =$$

$$\exp\left[-2\eta_1\eta_2\int_0^\infty \frac{dt}{t}\frac{\sinh\frac{\pi B}{4}\sinh\frac{\pi(2-B)}{4}}{\sinh t\cosh\frac{t}{2}}\cosh\left(t\left(n+i\frac{\theta_1-\theta_2}{\pi}-\frac{1}{2}(\eta_1-\eta_2)\right)\right)\right]$$

$$= \begin{cases} R(\theta_1-\theta_2;B,n) & \text{if } \eta_1 = \eta \\ \Phi(\theta_1-\theta_2;B,n)R(\theta_1-\theta_2;B,n) & \text{if } \eta_1 = -\eta_2 = 1 \\ \Phi(2\pi ni-\theta_1+\theta_2;B,n)R(\theta_1-\theta_2;B,n) & \text{if } \eta_1 = -\eta_2 = -1 \end{cases}$$

(138)

and consequently the FFs of the BPTF can be obtained as

$$\langle 0 \mid \mathcal{T}_n(0,0) \mid \theta_1,\dots\theta_k\rangle_{1\dots1} =$$

$$= \langle \mathcal{T}_n\rangle C^k C_\sigma^k \sum_{\eta_1,\dots,\eta_k} \exp\left[i\pi/2\sum_{l=1}^k \eta_l\right]\prod_{i<j}\langle\langle\Lambda_1^{\eta_i}(\theta_i+i\eta_i\pi/2)\Lambda_1^{\eta_j}(\theta_j+i\eta_j\pi/2)\rangle\rangle_\sigma,$$

(139)

where all the particles live on the first replica.

It is now easy to generalise the above construction to obtain the composite exponential branch point twist fields. We can retain and compute the average $\langle\langle\bullet\rangle\rangle_\sigma$ but choose the eigenvalue of $P$ as $p = g\alpha/(2\pi n)$ or equivalently use the projectors $\mathcal{P}(g\alpha/(2\pi n))$ to restrict to the corresponding Fock-module $\mathcal{F}_p$ instead of $\mathcal{F}_0$. We can again define the operators

$$Z_{\mu_j,\alpha}(\theta) = \mathcal{P}(g\alpha/(2\pi))Z_{\mu_j}(\theta)\mathcal{P}(g\alpha/(2\pi)),$$

$$Z_{\mu_j,\alpha}(\theta) = -iC\left[e^{i\pi\alpha B/(4\pi)}\Lambda_{\mu_j}^+(\theta+i\pi/2) - e^{-i\pi\alpha B/(4\pi)}\Lambda_{\mu_j}^-(\theta-i\pi/2)\right],$$

(140)

by which we can easily express any FF $F_k^\alpha$ of the composite field as

$$\langle 0 \mid \mathcal{T}_n^\alpha \mid \theta_1,\dots\theta_k\rangle_{1\dots1} = \langle\mathcal{T}_n^\alpha\rangle C^k C_\sigma^k\langle\langle Z_{1,\alpha/n}(\theta_1)\dots Z_{1,\alpha/n}(\theta_k)\rangle\rangle_\sigma$$

$$= \langle\mathcal{T}_n^\alpha\rangle C^k C_\sigma^k \sum_{\eta_1,\dots,\eta_k}\exp\left[(i\pi\alpha B/(4n\pi)-i\pi/2)\sum_{l=1}^k\eta_l\right]$$

$$\prod_{i<j}\langle\langle\Lambda_1^{\eta_i}(\theta_i+i\eta_i\pi/2)\Lambda_1^{\eta_j}(\theta_j+i\eta_j\pi/2)\rangle\rangle_\sigma,$$

(141)

when all the particles live on the 1st replica.

The square of the unknown constants $C, C_\sigma$ and can be unambiguously fixed by requiring the fulfilment of the kinematical pole equation

$$-i\operatorname*{Res}_{\theta=0}\tilde{F}_2(\theta+i\pi) = 1,$$

(142)

where $\tilde{F}_{bb}$ is any two-particle FF divided by the VEV of the corresponding field. We therefore have

$$C^2 = \frac{1}{\sin\frac{B\pi}{2}R(i\pi;B,1)},$$

$$C_\sigma^2 = \frac{\sin\frac{\pi}{2n}}{2n\sin\frac{\pi B}{4n}\sin\frac{\pi(2-B)}{4n}}\sin\frac{B\pi}{2},$$

(143)

from which we get the constant $\mathcal{C}(B, n) = C C_{\sigma}$ defined first in (81).

We can now easily compute the 1-,2-,3- and four-particle FFs of the composite field via Eq. (141). For the one-particle FF, we get

$$
\begin{aligned}
F_1^\alpha(B, n) &= \langle \mathcal{T}_n^\alpha \rangle \left( e^{i\pi \alpha B/(4n\pi) - i\pi/2} + e^{-i\pi \alpha B/(4n\pi) + i\pi/2} \right) \mathcal{C}(B, n) \\
&= \langle \mathcal{T}_n^\alpha \rangle 2 \sin \frac{\alpha B}{4n} \mathcal{C}(B, n).
\end{aligned}
\tag{144}
$$

For the two-particle FF, we have

$$
\begin{aligned}
F_2^\alpha(\theta; B, n) &= \langle \mathcal{T}_n^\alpha \rangle \mathcal{C}(B, n)^2 \left[ -2 \cos \frac{\alpha B}{2n} R(\theta; B, n) + \frac{1}{R(\theta - i\pi; B, n)} + \frac{1}{R(\theta + i\pi; B, n)} \right] \\
&= \langle \mathcal{T}_n^\alpha \rangle \mathcal{C}(B, n)^2 \left[ -2 + \Phi(2\pi n i - \theta; B, n) + \Phi(\theta; B, n) + \sin^2 \frac{\alpha B}{4n} \right] R(\theta; B, n),
\end{aligned}
\tag{145}
$$

from which (85) follows by analytic continuation.

For the three- and four-particle FFs, we obtain, as expected, rather complicated formulae:

$$
\begin{aligned}
F_3^\alpha(\theta_1, \theta_2, \theta_3; B, n) = \langle \mathcal{T}_n^\alpha \rangle \mathcal{C}(B, n)^3 \Bigg[ & \frac{e^{\frac{i\alpha B}{4n}} \mathcal{R}_2(\theta_{13}^+, \theta_{13}; B, n) - e^{-\frac{i\alpha B}{4n}} \mathcal{R}_2(\theta_{23}^-, \theta_{23}; B, n)}{i \mathcal{R}_3(\theta_{12}^+, \theta_{13}^+, \theta_{23}^-; B, n)} \\
& + \frac{e^{\frac{i\alpha B}{4n}} \mathcal{R}_2(\theta_{23}^+, \theta_{23}; B, n) - e^{-\frac{i\alpha B}{4n}} \mathcal{R}_2(\theta_{13}^-, \theta_{13}; B, n)}{i \mathcal{R}_3(\theta_{12}^-, \theta_{13}^-, \theta_{23}^+; B, n)} \\
+ \frac{i R(\theta_{12}; B, n) e^{-\frac{i\alpha B}{4n}}}{\mathcal{R}_2(\theta_{13}^-, \theta_{23}^-; B, n)} - & \frac{i R(\theta_{12}; B, n) e^{\frac{i\alpha B}{4n}}}{\mathcal{R}_2(\theta_{13}^+, \theta_{23}^+; B, n)} - 2 \sin \frac{3\alpha B}{4n} \mathcal{R}_3(\theta_{12}, \theta_{13}, \theta_{23}; B, n) \Bigg],
\end{aligned}
\tag{146}
$$

where $\theta_{ij} = \theta_i - \theta_j$, $\theta_{ij}^\pm := \theta_{ij} \pm i\pi$ and we introduced the slightly shorter notations

$$
\begin{aligned}
\mathcal{R}_2(\theta_1, \theta_2; B, n) &:= R(\theta_1; B, n) R(\theta_2; B, n), \\
\mathcal{R}_3(\theta_1, \theta_2, \theta_3; B, n) &:= R(\theta_1; B, n) R(\theta_2; B, n) R(\theta_3; B, n),
\end{aligned}
\tag{147}
$$

for the products of two and three $R$-functions. Also

$$
\begin{aligned}
F_4^\alpha(\theta_1, \theta_2, \theta_3, \theta_4; B, n) = \langle \mathcal{T}_n^\alpha \rangle \mathcal{C}(B, n)^4 \{ & \mathcal{R}_3(\theta_{12}, \theta_{13}, \theta_{23}; B, n) \\
\times \Bigg[ 2 \cos \frac{\alpha B}{n} \mathcal{R}_3(\theta_{14}, \theta_{24}, \theta_{34}; B, n) &- \frac{e^{-\frac{i\alpha B}{2n}}}{\mathcal{R}_3(\theta_{14}^-, \theta_{24}^-, \theta_{34}^-; B, n)} - \frac{e^{\frac{i\alpha B}{2n}}}{\mathcal{R}_3(\theta_{14}^+, \theta_{24}^+, \theta_{34}^+; B, n)} \Bigg]
\end{aligned}
$$

$$
\begin{aligned}
& + \frac{\frac{1}{\mathcal{D}(\theta_{14}^-, \theta_{24}^-, \theta_{34}; B, n)} - e^{-\frac{i\alpha B}{2n}} \mathcal{D}(\theta_{14}, \theta_{24}, \theta_{34}^+; B, n)}{\mathcal{D}(\theta_{13}^-, \theta_{23}^-, \theta_{12}; B, n)} + \frac{\frac{1}{\mathcal{D}(\theta_{14}^+, \theta_{24}^+, \theta_{34}; B, n)} - e^{\frac{i\alpha B}{2n}} \mathcal{D}(\theta_{14}, \theta_{24}, \theta_{34}^-; B, n)}{\mathcal{D}(\theta_{13}^+, \theta_{23}^+, \theta_{12}; B, n)} \\
& + \frac{\frac{1}{\mathcal{D}(\theta_{24}^-, \theta_{34}^-, \theta_{14}; B, n)} - e^{-\frac{i\alpha B}{2n}} \mathcal{D}(\theta_{24}, \theta_{34}, \theta_{14}^+; B, n)}{\mathcal{D}(\theta_{12}^+, \theta_{13}^+, \theta_{23}; B, n)} + \frac{\frac{1}{\mathcal{D}(\theta_{24}^+, \theta_{34}^+, \theta_{14}; B, n)} - e^{\frac{i\alpha B}{2n}} \mathcal{D}(\theta_{24}, \theta_{34}, \theta_{14}^-; B, n)}{\mathcal{D}(\theta_{12}^-, \theta_{13}^-, \theta_{23}; B, n)} \\
& + \frac{\frac{1}{\mathcal{D}(\theta_{14}^-, \theta_{34}^-, \theta_{24}; B, n)} - e^{-\frac{i\alpha B}{2n}} \mathcal{D}(\theta_{14}, \theta_{34}, \theta_{24}^+; B, n)}{\mathcal{D}(\theta_{12}^-, \theta_{23}^+, \theta_{13}; B, n)} + \frac{\frac{1}{\mathcal{D}(\theta_{14}^+, \theta_{34}^+, \theta_{24}; B, n)} - e^{\frac{i\alpha B}{2n}} \mathcal{D}(\theta_{14}, \theta_{34}, \theta_{24}^-; B, n)}{\mathcal{D}(\theta_{12}^+, \theta_{23}^-, \theta_{13}; B, n)} \Bigg\},
\end{aligned}
\tag{148}
$$

where

$$\mathcal{D}(\theta_1, \theta_2, \theta_3; B, n) = \frac{R(\theta_1; B, n)R(\theta_2; B, n)}{R(\theta_3; B, n)}, \tag{149}$$

and we recall that that $R(\theta^{\pm}; B, n)$ can be rewritten using $\Phi(\theta; B, n)$ of Eq. (130) as

$$
\begin{aligned}
R(\theta + i\pi; B, n)^{-1} &= \Phi(\theta; B, n)R(\theta; B, n,) \\
R(\theta - i\pi; B, n)^{-1} &= \Phi(2\pi n i - \theta; B, n)R(\theta; B, n).
\end{aligned} \tag{150}
$$

## B  $\Delta$-Sum Rule Checks

In this Appendix we provide some more tables complementing the results of Subsection 4.1 for additional parameter values. Recall that the exact conformal dimension of the CTF in the shG model is

$$\Delta_n^\alpha = \frac{1}{24}\left(n - \frac{1}{n}\right) + \frac{\Delta_1^\alpha}{n}, \qquad \text{with} \qquad \Delta_1^\alpha = -\frac{B(\alpha/(2\pi))^2}{2 - B} = -\frac{\alpha^2 g^2}{4(2\pi)^3}. \tag{151}$$

Recall that $\Delta_n^0$ is the dimension of the conventional BPTF.

Table 2: The $\Delta$-sum rule in the two-particle approximation (SR) compared with the exact conformal dimension of the exponential CFT in the shG model (61) for $\alpha_1 = 0.7039 \times 2\pi, \alpha_2 = 0.4483 \times 2\pi$ and $\alpha_3 = -0.5623 \times 2\pi$ and various values of $B$. For comparison, the tables include $\alpha = 0$, that is the standard BPTF and $n = 1$, $\alpha \neq 0$ corresponding to the exponential field in shG. In all cases the agreement is very good.

(a) $\alpha_1 = 0.7039 \times 2\pi, \alpha_2 = 0.4483 \times 2\pi, \alpha_3 = -0.5623 \times 2\pi, B = 0.2$

| $n$ | $\Delta_n^0$ (Exact) | $\Delta_n^0$ (SR) | $\Delta_n^{\alpha_1}$ (Exact) | $\Delta_n^{\alpha_1}$ (SR) | $\Delta_n^{\alpha_2}$ (Exact) | $\Delta_n^{\alpha_2}$ (SR) | $\Delta_n^{\alpha_3}$ (Exact) | $\Delta_n^{\alpha_3}$ (SR) |
|---|---|---|---|---|---|---|---|---|
| 1 | 0 | 0 | -0.055053 | -0.053919 | -0.022330 | -0.022084 | -0.086131 | -0.079370 |
| 2 | 0.0625 | 0.063569 | 0.034974 | 0.034953 | 0.051335 | 0.0519333 | 0.019434 | 0.018051 |
| 3 | 0.11111 | 0.113523 | 0.092760 | 0.094216 | 0.103668 | 0.105683 | 0.082401 | 0.083705 |
| 4 | 0.15625 | 0.159907 | 0.142487 | 0.145365 | 0.150667 | 0.154005 | 0.134717 | 0.138935 |
| 5 | 0.2 | 0.204842 | 0.188989 | 0.193184 | 0.195534 | 0.200112 | 0.182774 | 0.189689 |

(b) $\alpha_1 = 0.7039 \times 2\pi, \alpha_2 = 0.4483 \times 2\pi, \alpha_3 = -0.5623 \times 2\pi, B = 0.4$

| $n$ | $\Delta_n^0$ (Exact) | $\Delta_n^0$ (SR) | $\Delta_n^{\alpha_1}$ (Exact) | $\Delta_n^{\alpha_1}$ (SR) | $\Delta_n^{\alpha_2}$ (Exact) | $\Delta_n^{\alpha_2}$ (SR) | $\Delta_n^{\alpha_3}$ (Exact) | $\Delta_n^{\alpha_3}$ (SR) |
|---|---|---|---|---|---|---|---|---|
| 1 | 0 | 0 | -0.126843 | -0.117032 | -0.050243 | -0.048283 | -0.079045 | -0.074810 |
| 2 | 0.0625 | 0.064081 | -0.00092 | -0.00281 | 0.037378 | 0.037318 | 0.022977 | 0.022135 |
| 3 | 0.11111 | 0.114828 | 0.068830 | 0.069041 | 0.0943634 | 0.09661 | 0.084763 | 0.086215 |
| 4 | 0.15625 | 0.161949 | 0.124539 | 0.127284 | 0.143689 | 0.148183 | 0.136489 | 0.140313 |
| 5 | 0.2 | 0.207582 | 0.174631 | 0.179727 | 0.189951 | 0.196531 | 0.184191 | 0.190206 |

(c) $\alpha_1 = 0.7039 \times 2\pi, \alpha_2 = 0.4483 \times 2\pi, \alpha_3 = -0.5623 \times 2\pi, B = 0.6$

| $n$ | $\Delta_n^0$ (Exact) | $\Delta_n^0$ (SR) | $\Delta_n^{\alpha_1}$ (Exact) | $\Delta_n^{\alpha_1}$ (SR) | $\Delta_n^{\alpha_2}$ (Exact) | $\Delta_n^{\alpha_2}$ (SR) | $\Delta_n^{\alpha_3}$ (Exact) | $\Delta_n^{\alpha_3}$ (SR) |
|---|---|---|---|---|---|---|---|---|
| 1 | 0 | 0 | -0.217445 | -0.182639 | -0.086131 | -0.079370 | -0.135506 | -0.120618 |
| 2 | 0.0625 | 0.064306 | -0.046222 | -0.049837 | 0.019435 | 0.018051 | -0.0052531 | -0.007845 |
| 3 | 0.11111 | 0.115505 | 0.038630 | 0.036031 | 0.082401 | 0.083705 | 0.065942 | 0.065666 |
| 4 | 0.15625 | 0.163049 | 0.101889 | 0.102516 | 0.134717 | 0.138935 | 0.122373 | 0.125193 |
| 5 | 0.2 | 0.20908 | 0.156511 | 0.160304 | 0.182774 | 0.189689 | 0.172899 | 0.178615 |

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
