# Peer review of "Branch Point Twist Field Form Factors in the sine-Gordon Model II: Composite Twist Fields and Symmetry Resolved Entanglement"

_SciPost Physics, doi:SciPost Phys. 12, 088 (2022)_

## Round 2 · Referee Report · Anonymous (Referee 1) · 2021-10-21

Strengths

1-The paper extends previous mathematical methods.
2-The subject (symmetry resolved entanglement) is a subject of current interest.
3-The paper is clear and well written

Report

This is a natural continuation of a previous paper of two of the authors (ref
22), where several entanglement measures are calculate for the sine-Gordon
field theory. In the previous calculations the authors does not split the
particular contribution for the several U(1) symmetry sectors of the model.
This is done in this paper. For this sake, they have to consider partition
functions arising not only from the normal twist fields (due to replica
trick), but composed to the vertex U(1) operators. They extend the previous resu
lt for the form factors of composite twist fields to extend to the
non-diagonal case (composite to the vertex operator, most probably is
interesting for further studies.

The paper makes a clear explanation of the model and physics behind, it is
well written and I suggest its acceptance in the journal.

---

## Round 2 · Referee Report · Anonymous (Referee 2) · 2021-10-30

Strengths

1- computes a quantity of interest (symmetry-resolved entanglement entropies) in a new physical setting 2- high level of technicality, but the presentation remains clear and reasonably self-consistent

Weaknesses

1- while context is given on twist fields and the SG/ShG theories, the introduction of symmetry resolved entropies is less motivated here.

Report

This is a technically rich, but very clearly written work, where the authors compute, to leading order, the symmetry resolved entanglement entropies (SREE) in the attractive Sine-Gordon quantum field theory. This theory has the features of being interacting, with a U(1) symmetry and non-diagonal scattering, and it is the first time SREEs are computed in such a setting.
In a former paper by two of the authors the entanglement entropies were computed, involving the study of branch point twist fields. The novelty here is that to study SREEs those twist fields have to be composed with other twist fields associated with the U(1) symmetry (the vertex operators) . The corresponding form factors in the breather sector are computed by resolution of the bootstrap equations, and successfully compared to the "$\Delta$-sum rule". From there, the SREE are computed to leading order in the UV cutoff, displaying the property of equipartition.
I think this is a very relevant work, technically consequent yet clearly presented, which computes a quantity of timely and widespread interest in a physically relevant and new setting. It also constitutes an important step in the programme of generally computing SREEs in non-diagonal integrable QFTs (the next step of tackling sectors with solitons and antisolitons being another challenging issue, as mentioned by the authors in the conclusion).
There are very few minor aspects that I'd like to point out (see next section), but besides those I recommend this article for publication in SciPost.

Requested changes

1- around eq (61) : it might seem confusing to the reader to read about "conformal dimension", for a theory which is not a CFT... It would probably be useful to add, there or when introducing the SG/ShG models, an explanation about their construction as perturbed CFTs

2-in the end of the work, the authors conclude about equipartition at leading order in the UV cutoff $\epsilon$. Could they comment more on equipartition breaking at lower orders ? It seemed to me that equipartition was a general feature observed in all models.

3- there are a few typos or grammar mistakes. I spotted some but there are certainly a few more. for instance : -last paragraph of p.2 : "dubbed as"- > "dubbed", "similarly" -> similar - eq (25) there's one too many ":" signs in the normal ordering

  • validity: top
  • significance: high
  • originality: high
  • clarity: top
  • formatting: excellent
  • grammar: excellent

Author:  David Horvath  on 2021-11-16  [id 1947]

(in reply to Report 2 on 2021-10-30)

Answer Referee Report on “Branch Point Twist Field Form Factors in the sine-Gordon Model II: Composite Twist Fields and Symmetry Resolved Entanglement”

We would like to thank both referees for their positive comments on our work.
Referee two raised three points that we address below:

-around eq (61): it might seem confusing to the reader to read about "conformal dimension", for a theory which is not a CFT... It would probably be useful to add, there or when introducing the SG/ShG models, an explanation about their construction as perturbed CFTs

We understand why the confusion may arise, although it is perhaps worth saying that people working in this field are very accustomed to recovering CFT data while studying perturbations of CFT or using such CFT data as benchmarks for checking form factor solutions (as in the Delta-sum rule). As the referee suggests, we have added two sentences in the introduction (after equation (10) and after equation (22)) to indicate that both the sinh-Gordon and sine-Gordon models may be seen as massive perturbations of a massless free boson and therefore their underlying CFT or UV limit is a free CFT model with central charge c=1. 

-in the end of the work, the authors conclude about equipartition at leading order in the UV cutoff ϵ. Could they comment more on equipartition breaking at lower orders ? It seemed to me that equipartition was a general feature observed in all models.

According to the referee’s request, we have added three more formulae (Eq. 110, 113 and 116 in the new version), from which Eqs. 113 and 116 explicitly display the leading order breaking of equipartition. In addition, the leading (non-trivial) q-dependent term was not correctly identified in Eq. 109, which we have corrected as well. Although relatively straightforward, we have also added some explanatory text to summarise how these contributions can be obtained.

-there are a few typos or grammar mistakes. I spotted some but there are certainly a few more. for instance :

last paragraph of p.2 : "dubbed as"- > "dubbed", "similarly" -> similar 
- eq (25) there's one too many ":" signs in the normal ordering

We have corrected all of these and a few others we found.

In addition, we have realised that there was a typo in our equation (25) where the conformal dimension \Delta_n should have been \Delta, the dimension of the field \phi. The referee’s comment on having too many normal orderings in this equation stems from a misreading of the equation. Whereas the first two “:” indicate normal ordering, the last one “:=” means that this is a definition of the composite field. We have introduced an extra space to hopefully make this clearer.

List of changes

last paragraph of p.2 : "dubbed as"- > "dubbed", "similarly" -> similar
last paragraph of p.3 : "similarly" -> similar
we have added two sentences in the introduction (after equation (10) and after equation (22))
p.6 under Eq. 17 : "similarly" -> similar
p. 7 under Eq. 25: "conformal dimension of the BPTF" -> "conformal dimension of the field φ, denoted by ∆. The conformal dimension of the BPTF is given by"
p. 20-21: a correction in Eq. 109 (including the more precise specification of subleading terms) and below the insertion of Eq. 110 and 3 sentences explaining the extraction of the q-dependent corrections
p. 21: the insertion of Eq. 113 in a new explanatory sentence and the addition of Eq. 116 with some words of comment too.
Updating the reference list by adding the journal reference to papers that have been published but were preprints when our submission occurred.

---

## Round 2 · List of Changes

last paragraph of p.2 : "dubbed as"- > "dubbed", "similarly" -> similar
last paragraph of p.3 : "similarly" -> similar
we have added two sentences in the introduction (after equation (10) and after equation (22))
p.6 under Eq. 17 : "similarly" -> similar
p. 7 under Eq. 25: "conformal dimension of the BPTF" -> "conformal dimension of the field φ, denoted by ∆. The conformal dimension of the BPTF is given by"
p. 20-21: a correction in Eq. 109 (including the more precise specification of subleading terms) and below the insertion of Eq. 110 and 3 sentences explaining the extraction of the q-dependent corrections
p. 21: the insertion of Eq. 113 in a new explanatory sentence and the addition of Eq. 116 with some words of comment too.
Updating the reference list by adding the journal reference to papers that have been published but were preprints when our submission occurred.

Report #1 by Anonymous (Referee 1) on 2021-10-21 (Invited Report)
Strengths

1-The paper extends previous mathematical methods.
2-The subject (symmetry resolved entanglement) is a subject of current interest.
3-The paper is clear and well written

Report

This is a natural continuation of a previous paper of two of the authors (ref
22), where several entanglement measures are calculate for the sine-Gordon
field theory. In the previous calculations the authors does not split the
particular contribution for the several U(1) symmetry sectors of the model.
This is done in this paper. For this sake, they have to consider partition
functions arising not only from the normal twist fields (due to replica
trick), but composed to the vertex U(1) operators. They extend the previous resu
lt for the form factors of composite twist fields to extend to the
non-diagonal case (composite to the vertex operator, most probably is
interesting for further studies.

The paper makes a clear explanation of the model and physics behind, it is
well written and I suggest its acceptance in the journal.
  • validity: high
  • significance: good
  • originality: good
  • clarity: high
  • formatting: excellent
  • grammar: good

---

## Round 3 · Referee Report · Anonymous (Referee 2) · 2021-11-16

Report

I thanks the authors for their additions to the manuscript, which I recommend for publication in SciPost in its present form.

---

## Editorial Decision

published